# Theoretically Motivated Dark Electromagnetism as the Origin of Relativistic Modified Newtonian Dynamics

Felix Finster [1], José M. Isidro [2] , Claudio F. Paganini [1] and Tejinder P. Singh [3,4,*]

1   Fakultät für Mathematik, Universität Regensburg, D-93040 Regensburg, Germany; finster@ur.de (F.F.); claudio.paganini@mathematik.uni-regensburg.de (C.F.P.)
2   Instituto Universitario de Matemática Pura y Aplicada, Universitat Politècnica de València, 46022 Valencia, Spain; joissan@mat.upv.es
3   Inter-University Centre for Astronomy and Astrophysics, Post Bag 4, Ganeshkhind, Pune 411007, India
4   Tata Institute of Fundamental Research, Homi Bhabha Road, Mumbai 400005, India
*   Correspondence: tejinder.singh@iucaa.in or tpsingh@tifr.res.in

**Abstract:** The present paper is a modest attempt to initiate the research program outlined in this abstract. We propose that general relativity and relativistic MOND (RelMOND) are analogues of broken electroweak symmetry. That is, $SU(2)_R \times U(1)_{YDEM} \to U(1)_{DEM}$ (DEM stands for dark electromagnetism), and GR is assumed to arise from the broken $SU(2)_R$ symmetry and is analogous to the weak force. RelMOND is identified with dark electromagnetism $U(1)_{DEM}$, which is the remaining unbroken symmetry after the spontaneous symmetry breaking of the dark electro-grav sector $SU(2)_R \times U(1)_{YDEM}$. This sector, as well as the electroweak sector, arises from the breaking of an $E_8 \times E_8$ symmetry in a recently proposed model of unification of the standard model with pre-gravitation, with the latter based on an $SU(2)_R$ gauge theory. The source charge for the dark electromagnetic force is the square root of mass, motivated by the experimental fact that the ratio of the square roots of the masses of the electron, up-quark, and down-quark is 1:2:3, which is the opposite of the ratio of their electric charges at 3:2:1. The introduction of the dark electromagnetic force helps us understand the peculiar mass ratios of the second and third generations of charged fermions. We also note that in the deep MOND regime, acceleration is proportional to the square root of mass, which motivates us to propose the relativistic $U(1)_{DEM}$ gauge symmetry as the origin of MOND. We explain why the dark electromagnetic force falls inversely with distance, as in MOND, rather than following the inverse square of distance. We conclude that dark electromagnetism effectively mimics cold dark matter, and the two are essentially indistinguishable in cosmological situations where CDM successfully explains observations, such as CMB anisotropies and gravitational lensing.

**Keywords:** dark electromagnetism; dark photon; unification; dark matter; MOND; emergent gravity



## 1. Introduction

The cold dark matter paradigm is a cornerstone of modern cosmology. Dark matter plays a central role in explaining structure formation in the early Universe, as well as the observed anisotropies of the cosmic microwave background. Dark matter is also pivotal in explaining the gravitational lensing of large-scale structures and the observed baryon acoustic oscillations. Anomalous velocity dispersions in galactic clusters are also accounted for by proposing the existence of dark matter, and while facing some challenges, galaxy rotation curves can also be explained by assuming a specific density profile for dark matter. Thus, a single assumption, that of a prevailing collisionless and gravitationally attractive fluid with a mass density about 5–6 times that of ordinary matter, can account for a wide range of cosmological phenomena. A new elementary particle, which is beyond the standard model of particle physics, is considered the most likely origin of this dark fluid.

And yet, the greatest challenge for dark matter is that no such particle has been experimentally detected in the laboratory so far, despite a very large number of proposed

theoretical candidates, as well as intense experimental searches over some four decades and through nearly fifty different experiments. Of course, this does not imply that such a particle will never be detected, but it does raise the question of whether there could be an alternative explanation of cosmic phenomena in which dark matter is not required. Instead, one might consider the possibility that the law of gravitation (both Newton's law and general relativity) could be modified on large scales. It is not impossible that such a modification to gravitation perfectly mimics the proposed dark fluid. Thus, what we call dark matter could, in fact, be modified gravity in disguise. It is this idea that is explored in the present paper, starting from a first-principles proposal for modified gravity, which we call dark electromagnetism.

The most successful proposal for a modified law of gravitation as an alternative to dark matter is Milgrom's Modified Newtonian Dynamics (MOND). This is a phenomenological proposal in which, for accelerations far smaller than a certain low critical value, the law of gravitation (in the non-relativistic limit) changes from an inverse square to a $1/R$ law. MOND does very well at explaining galaxy rotation curves, whereas on cluster scales and cosmic scales, it has modest success and also faces serious challenges. MOND's major challenge is that it lacks a first-principles theoretical explanation: What are the convincing theoretical reasons (independent of observations) that might compel us to modify general relativity in such a way that MOND is an inevitable consequence of the modified theory in its non-relativistic limit?

Our recent proposal for a unified theory of interactions provides strong evidence for a theoretical origin of MOND in the form of a fifth force. Surprisingly, this unified theory does not provide any dark matter particle candidate, thus favouring modified gravity over dark matter. This prediction also makes the theory eminently falsifiable—it can be ruled out through laboratory detection of dark matter. The present paper describes the origin of this fifth force (dark electromagnetism) and how it serves as the origin of MOND. The theory is briefly reviewed in Section 2, followed by a brief review of MOND in Section 3.

As we explain in Section 6 of this paper, our proposal for MOND as dark electromagnetism also explains Verlinde's entropic criterion [1] for MOND. Keeping this connection with Verlinde in mind, in Section 4, we review Verlinde's proposal for motivating MOND from entropy considerations.

Section 5 is central to this paper, as it explains how dark electromagnetism mimics MOND on various cosmic scales. Related empirical aspects of dark electromagnetism are discussed in Section 7. Section 8 critiques our findings, and we also comment on the current challenges faced by MOND.

## 2. Theoretical Origin of Dark Electromagnetism

A theory of unification of the fundamental forces has recently been proposed [2], starting from the foundational requirement that there should exist a reformulation of quantum field theory that does not depend on classical time [3]. This theory is based on an $E_8 \times E_8$ symmetry group, in which each of the two $E_8$ groups is assumed to branch as follows as a result of spontaneous symmetry breaking, which is identified with the electroweak symmetry breaking:

$$E_8 \longrightarrow SU(3) \times E_6 \longrightarrow \quad SU(3) \times SU(3) \times SU(3) \times SU(3) \longrightarrow \tag{1}$$

$$SU(3) \times SU(3) \times SU(3) \times SU(2) \times U(1) \tag{2}$$

Leaving the two $E_6$s aside for a moment, the $SU(3) \times SU(3)$ pair arising from the $E_8 \times E_8$ branching is mapped to an (8+8=16)-dimensional split bioctonionic space from which our 4D spacetime, as well as the internal symmetry space for the standard model forces (and two newly predicted forces), is assumed to emerge.

The three $SU(3)$s arising from the branching of each of the two $E_6$s, with the rightmost $SU(3)$ in each set branching as $SU(2) \times U(1)$, are interpreted as follows. In the first $E_6$, the first $SU(3)$ is $SU(3)_{genL}$ and describes three generations of left-handed standard model

fermions (eight per generation, along with their anti-particles). The second $SU(3)$ is associated with the $SU(3)_{color}$ of QCD. The branched third $SU(3) \rightarrow SU(2)_L \times U(1)_Y$ describes the electroweak symmetry of the standard model, broken into $U(1)_{em}$.

In the second of the two $E_6$s, the first $SU(3)$ is $SU(3)_{genR}$ and describes three generations of standard model right-handed fermions, including three types of sterile neutrinos (eight fermions per generation, along with their anti-particles). The second $SU(3)$ is identified with a newly predicted but yet-to-be-discovered new (likely short-range) 'sixth force' named $SU(3)_{grav}$. The third $SU(3) \rightarrow SU(2)_R \times U(1)_{YDEM}$ describes what we call the dark electro-grav sector, which breaks into the newly predicted 'fifth force' $U(1)_{DEM}$, which we name dark electromagnetism. We propose this to be the relativistic MOND theory (a gauge theory), whose non-relativistic limit is Milgrom's MOND [4]. The broken $SU(2)_R$ symmetry is proposed to give rise to classical gravitation described by the general theory of relativity (GR). At low accelerations, the fifth force of dark electromagnetism (DEM) dominates over GR, whereas at high accelerations, GR dominates over DEM, with the transition coming at the critical MOND acceleration $a_M \sim a_0/6 \approx cH_0/6$, where $a_0$ is the cosmological acceleration of the current accelerating Universe. We reiterate that standard general relativity is assumed to emerge from the broken $SU(2)_R$ symmetry, whereas $SU(3)_{grav}$ is a newly predicted unbroken symmetry (likely short-range and extremely weak, and in which the charged leptons and down family of quarks take part).

The particle content of this unification proposal has been described in detail by Kaushik et al. [2]. All the $248 + 248 = 496$ degrees of freedom of $E_8 \times E_8$ are accounted for. The only fermions in the theory are three generations of standard model chiral fermions. Apart from the 12 standard model gauge bosons, there are 12 newly predicted spin-one gauge bosons associated with the $SU(3)_{grav} \times SU(2)_R \times U(1)_{YDEM}$ sector. Eight of these are so-called gravi-gluons associated with the (likely to be short-range and ultra-weak compared to QCD) $SU(3)_{grav}$. The gauge boson associated with $U(1)_{DEM}$ is named the dark photon. It is massless and has zero electric charge. Of the three bosons associated with the broken $SU(2)_R$ symmetry, two have zero electric charges but are as massive as the Planck mass, and hence mediate interactions at the Planck length range: these are analogues of the $W^+$ and $W^-$ bosons of the weak force. The third is massless and has an insignificantly tiny electric charge (scaled down enormously due to cosmological inflation in comparison to the charge of the electron), which can be set to zero for all practical purposes. This boson is the analogue of the $Z^0$ of the weak force. The pre-gravitation $SU(2)_R$ symmetry is mediated by spin-one gauge bosons, with gravitation, as described by the metric tensor in the general theory of relativity, emerging only in the classical limit. In our approach, classical GR is not to be quantised, which is why we do not have a fundamental, non-composite, spin-2 graviton in the theory. This does not contradict the fact that classical GR admits the experimentally confirmed quadrupolar gravitational waves. The apparent spin-2 nature of gravitation is emergent only in the classical limit. The underlying theory from which spacetime and GR emerge in the classical limit is a pre-quantum, pre-spacetime theory. Gravitation and quantum theory are emergent phenomena.

There are two Higgs doublets in this theory. The first is a standard model Higgs, which gives mass to left-chiral fermions upon spontaneous breaking of the electroweak symmetry. The second is a newly predicted Higgs boson, which gives electric charge to the right-chiral fermions upon breaking of the dark electro-grav symmetry and coincides with the electroweak symmetry breaking. Unlike in the standard model, both Higgs are now predicted to be composite, being composed of the very fermions to which they give mass and electric charge. Of the 496 degrees of freedom in the theory, 32 are with the bosons (after including 4 each for the two Higgs). Moreover, 32 degrees of freedom are with the internal generation space and pre-spacetime (16 each), and 144 d.o.f. are with the fermions. The remaining 288 d.o.f. go into making two composite Higgs, 144 per Higgs. It is noteworthy that each Higgs has as many composite d.o.f. as the total number of d.o.f. in the fermions. The bosonic content of the theory can also be confirmed by examining the Lagrangian of the theory, as done by Raj and Singh [5].

The source charge associated with $U(1)_{em}$ is, of course, the electric charge, and in the algebraic approach to unification, it can be shown to be quantised, as done, for instance, by Furey [6]. The electric charge is defined as the number operator constructed from generators of the Clifford algebra $Cl(6)$, which, in turn, is generated by octonionic chains acting on octonions. It has been shown that the electric charge can only take the values $(0, 1/3, 2/3, 1)$. Furthermore, the spinorial states associated with these charge values exhibit the following symmetry under the group $SU(3)$ (which is a maximal subgroup of the smallest exceptional group $G_2$, the automorphism group of the octonions). States with charges 0 and 1 are singlets of $SU(3)$, states with charge 1/3 are anti-triplets, and states with charge 2/3 are triplets. This enables the interpretation that states with charges (0, 1/3, 2/3, 1) are, respectively, the (left-handed) neutrino, anti-down-quark, up-quark, and positron. The SU(3) is hence identified with the $SU(3)_{color}$ of QCD. Anti-particle states are obtained by complex conjugation of particle states and are shown to have opposite signs of electric charge, as anticipated. Note that these fermions are left-chiral particles, and their corresponding anti-particles are right-chiral. Furthermore, this quantisation of electric charge holds for every one of the three fermion generations. The Clifford algebra construction applies equally well to the second and third generations.

Next, consider the symmetry $SU(3)_{grav} \times U(1)_{DEM}$ associated with the right-handed sector, with these two being the two new forces [2]. Now, the source charge associated with the $U(1)_{DEM}$ symmetry is the square root of mass $\pm\sqrt{m}$, not the electric charge. The motivation for proposing this interpretation (for the number operator made from the Clifford algebra Cl(6) generators that define the right-chiral fermions) comes from the following remarkable fact [7]. The ratio of the square roots of the masses of the electron, up-quark, and down-quark is 1:2:3, which is the opposite of the ratios of their electric charges ratio at 3:2:1. We treat the electric charge and the square root of mass on the same footing. The square root of mass also takes two signs: $+\sqrt{m}$ and $-\sqrt{m}$. The positive sign is for matter, and the negative sign is for anti-matter: like signs attract under dark electromagnetic force and unlike signs repel. Note that mass $m$, being the square of $\pm\sqrt{m}$, is necessarily positive. Three new colours for $SU(3)_{grav}$ are introduced: the right-handed neutrino and the down-quark are singlets of these new colours and have $\sqrt{m}$ values of 0 and 1, respectively. The electron is an anti-triplet of $SU(3)_{grav}$ with an $\sqrt{m}$ value of 1/3, and the up-quark is a triplet of $SU(3)_{grav}$ with an $\sqrt{m}$ value of 2/3. Their anti-particles have corresponding square-root mass values $-\sqrt{m}$. This mass quantisation is derived from first principles, just as for the electric charge quantisation, and holds for each of the three generations, just like for the electric charge. Note that our proposal also gives a dynamical definition for matter/anti-matter: matter has a positive sign of the square root of mass $(+\sqrt{m})$, whereas anti-matter has a negative sign of the square root of mass $(-\sqrt{m})$. Mass $m$ is, of course, positive for both matter and anti-matter and is obtained from squaring $\pm\sqrt{m}$.

Why, then, do the second and third fermion generations have such peculiar mass ratios, as observed in experiments [8]? The answer is that even when we conduct experiments to measure particle masses, the measurements are electromagnetic in nature and based on electric charge eigenstates. However, these electric charge eigenstates do not correspond to the (square-root) mass eigenstates. The exceptional Jordan algebra associated with the three fermion generations (one algebra for electric charge eigenstates that are left-chiral, and one algebra for the square-root mass eigenstates that are right-chiral) can be used to express the electric charge eigenstates as superpositions of the square-root mass eigenstates using the so-called Jordan eigenvalues. The weights of these superpositions reveal the observed mass ratios with very good accuracy [9], strongly supporting the proposal that the source charge associated with the dark electromagnetic force is the square root of mass. The fact that the source charge for the MOND acceleration is also the square root of mass encourages us to identify dark electromagnetism with relativistic MOND.

In the very early Universe, at the epoch of electroweak symmetry breaking, the enormous repulsive dark electromagnetic force segregated matter $(+\sqrt{m})$ from anti-matter $(-\sqrt{m})$ so that our part of the matter–anti-matter symmetric Universe is matter-dominated [5]. This

scenario bears resemblance to the CPT symmetric universe model proposed by Boyle and Turok [10–12]. As a result, the dark electromagnetic force in our matter-dominated Universe is apparently attractive only, even though $U(1)_{em}$ is a vector interaction). Similarly, the emergent gravitational interaction, which is the classical limit of the $SU(2)_R$ gauge theory, is attractive only. We predict that the $U(1)_{DEM}$ force between an electron and a positron is repulsive.

Another important aspect of the octonionic theory [3] (i.e., the one based on $E_8 \times E_8$ symmetry) is the 'square root of spacetime'. The spinorial states, which define the fermions and satisfy the Dirac equation, are constructed from the algebra of the octonions acting on themselves. In this sense, a spinor is the square of an octonion, and since spinors are defined on spacetime, this suggests the view that a space that is labelled using octonions as coordinates is actually the square root of spacetime. However, the absolute square modulus of an octonion should be assigned dimensions of length-squared, not length (as the Lagrangian of the octonionic theory suggests). This compels us to introduce the effective distance $R_{eff}^2 = R_H R$ in the unbroken theory, where $R_H$ is the de Sitter horizon. An unbroken symmetry such as $U(1)_{DEM}$ thus ought to have a distance dependence (say in Coulomb's law) as $1/(\sqrt{R_{eff}})^2 \sim 1/R$, and not $1/R^2$. This is a possible explanation for the $1/R$ dependence of the MOND acceleration, which, taken together with the source charge for $U(1)_{DEM}$, can explain why the MOND acceleration behaves as $\sqrt{M}/R$, unlike gravitation, which goes as $M/R^2$ in the Newtonian limit. We can say that gravitation is the square of dark electromagnetism: the source current for DEM is $\sqrt{m}cu^i$, whereas the source current for gravitation is $(\sqrt{m}cu^i)(\sqrt{m}cu^j) = mc^2u^iu^j$, which is nothing but the energy-momentum tensor.

In our proposal for DEM as relativistic MOND, the DEM force mimics Maxwell's electrodynamics, with the electric charge replaced with the square root of mass, and the spatial distance replaced with an effective distance $R_{eff} \equiv \sqrt{RR_H}$, where $R_H$ is the Hubble radius, or equivalently, the de Sitter horizon. The source term, in the non-relativistic limit, is the effective volume density of square-root mass: $\sqrt{M}/R_{eff}^3$. The left-hand side of the Poisson equation is the Laplacian made using the effective distance function. Such a Poisson equation yields MOND in the deep MOND regime.

Since MOND has a distance dependence in the acceleration as $1/R$, the associated MOND potential is logarithmic. This is, in principle, consistent with the source being a surface square-root mass density $\sim \sqrt{M}/R^2$, as if the MOND dynamics were taking place effectively in two spatial dimensions. This is consistent with the logarithmic form for the Green function of the Laplace equation in 2D, as we review in Appendix A. However, it should be noted that this surface density of square-root mass does not have a well-defined limit as $R \to 0$ (it diverges as $R^{-1/2}$), whereas the volume density of square-root mass, defined using the effective distance, does have a well-defined limit that is finite. Also, we would like to maintain the structure of the left-hand side of the Poisson equation, and this is consistent with proposing MOND as the non-relativistic limit of the $U(1)_{DEM}$ gauge theory.

Why do we associate the $SU(2)_R$ gauge symmetry with gravitation, as in the general theory of relativity? The following arguments provide a number of independent hints in favour of the notion that the group $SU(2)_R$ (arising in the octonionic theory [2,3,5]) qualifies to describe a theory of gravity in four dimensions.

In the octonionic theory, the product group $SU(2)_L \times SU(2)_R$ emerges, with one copy being left-handed and the other being right-handed. Now, $SU(2)_L \times SU(2)_R$ is locally isomorphic to $SO(4)$, the rotation group in four Euclidean dimensions. A Wick rotation will transform $SO(4)$ into the Lorentz group $SO(1,3)$. So, the Lorentz group in four dimensions is locally isomorphic to the product group $SU(2)_L \times SU(2)_R$.

That the left-handed subgroup $SU(2)_L$ accounts for the weak interaction within the standard model has been known for a long time. Here, we claim that the right-handed subgroup $SU(2)_R$ can account for gravity in four dimensions.

To see how a graviton could possibly emerge in this setting, consider the tensor product $\mathbf{1} \otimes \mathbf{1}$ of two copies of the three-dimensional irreducible representations of $SU(2)$.

Now, $\mathbf{1} \otimes \mathbf{1} = \mathbf{2} \oplus \mathbf{1} \oplus \mathbf{0}$. The $\mathbf{2}$ representation carries spin 2 and can thus accommodate the graviton. We expect the $\mathbf{2}$ representation to accommodate the emergent spin-2 graviton, with the $\mathbf{1}$ representation being the gravitational analogue of the electroweak $W^{\pm}$ and $Z^0$. The $\mathbf{0}$ representation might correspond to the standard model Higgs.

Moreover, Fermi's phenomenological theory of weak interactions has a Lagrangian that carries the Fermi constant $G_F$ multiplying the product of two currents; the dimension of $G_F$ is [energy]$^{-2}$. On the other hand, general relativity has a Lagrangian that carries Newton's constant $G_N$, where the coupling constant is the inverse $1/G_N$. Incidentally, the dimension of $G_N$ is again [energy]$^{-2}$. However, $G_N$ appears in the denominator within its Lagrangian, as opposed to $G_F$, which appears in the numerator.

That both $G_F$ and $G_N$ are dimensionful makes the corresponding theories nonrenormalisable. However, since both are effective theories representing low-energy limits of more fundamental theories, nonrenormalisability is not an issue.

All these hints make one suspect that gravity and the weak force could share a common origin, namely the group $SU(2)_L \otimes SU(2)_R$ within the octonionic theory. That the product of the two coupling constants $G_F$ and $1/G_N$ is *dimensionless* suggests the intriguing possibility that Fermi's theory and general relativity might be each other's dual under a $Z_2$ duality transformation exchanging the weak and strong coupling regimes. This duality is strongly reminiscent of the analogous dualities put forward in the literature [13,14].

Altogether, this suggests that one can consider $SU(2)_L$ as the dual of the theory governed by $SU(2)_R$. Gravity would then appear as the weak dual of the Fermi theory, with the latter being its strong counterpart. Mention should also be made of the various attempts made in the past towards gravi-weak unification [15].

Further evidence of a possible connection between the $SU(2)_R$ gauge symmetry and gravity comes from the work of Ashtekar [16], Krasnov [17], and Woit [18,19]. There is also the appealing fact that $SU(2)_R \times U(1)_{YDEM}$ (i.e., dark electro-grav) is a renormalisable gauge theory, similar to the electroweak theory $SU(2)_L \times U(1)_Y$.

The cosmological setting for our proposal of dark electromagnetism is as follows [20]. Subsequent to the Big Bang creation event, the Universe undergoes an inflation-like expansion. The expansion begins with a Planck-scale acceleration of $\sim 10^{53}$ cm s$^{-2}$, with the acceleration decreasing inversely with the expanding scale factor. One input taken from observations is that the Universe has $N \sim 10^{80}$ particles and hence a total mass of about $10^{55}$ g. The inflating epoch undergoes a phase transition, where the decreasing acceleration equals the surface gravity of a black hole with the same mass as that of the observed Universe. This acceleration happens to be of the same order as the critical MOND acceleration of $\sim 10^{-8}$ cm s$^{-2}$, as well as the acceleration of the current Universe. Hence, there is an inflation of the scale factor by 61 orders of magnitude before the inflation-like phase ends. (Incidentally, this inflation by 61 orders of magnitude reduces the cosmological constant— dimensionally inverse squared length—by 122 orders of magnitude, to the same order as its currently observed value). This phase transition also signifies a quantum-to-classical transition, and because the black hole surface gravity is now higher than the inflationary acceleration, classical inhomogeneous structures begin to form, and classical spacetime, obeying the laws of general relativity, emerges. This transition also embodies the electroweak symmetry breaking and the dark electro-grav symmetry breaking. Near compact objects, GR, emerging from the broken $SU(2)_R$ symmetry, dominates, whereas far from compact objects (once the induced acceleration falls below the critical MOND acceleration) the unbroken symmetry $U(1)_{DEM}$ of dark electromagnetism dominates. This latter scenario represents the deep MOND regime. Thus, in the presence of compact objects, the de Sitter horizon does not immediately yield to GR; rather the MOND zone mediates between the GR zone and the horizon. It is as if there is a phase transition between the GR zone and the MOND zone (similar to Verlinde's ideas [1]). This could potentially be elucidated through a generalisation of 'GR as thermodynamics' to '(GR + MOND) as thermodynamics' of an unbroken symmetry phase transforming into a broken symmetry phase. The GR-dominated phase manifests as the broken symmetry phase and is stiff, whereas the MOND phase

represents the unbroken symmetry phase and is elastic—the deep MOND region carries a memory of the unbroken inflation-like phase, as well as the currently accelerating Universe.

We note that grand unified theory (GUT) models based on $E_6$ symmetry have been previously considered by several researchers [21–23], and the significance of $E_6$ has been noted repeatedly (it is the only exceptional Lie group that has complex representations). Our proposal, the octonionic theory, is not a GUT. We propose an $E_6 \times E_6$ unification of standard model forces with gravitation, and we predict two new forces, $SU(3)_{grav}$ and $U(1)_{DEM}$. The inflation-like expansion resets the scale of quantum gravity from the Planck scale to the scale of electroweak symmetry breaking, i.e., $\sim 1$ TeV. This is also the scale of the breaking of the dark electro-grav symmetry $SU(2)_R \times U(1)_{YDEM}$, when spacetime and gravitation emerge from the pre-quantum, pre-spacetime theory. Relativistic MOND $U(1)_{DEM}$ also emerges at this epoch.

The term dark electromagnetism/dark radiation/dark photon is sometimes used to refer to a hypothetical radiation that mediates interactions between dark matter particles. In our proposal, however, this dark radiation mediates a fifth force between ordinary baryonic matter particles (and, of course, between leptons as well). There is no dark matter in our theory unless one wishes to refer to the dark photons of DEM as dark matter.

### 3. A Brief Review of MOND and Relativistic MOND

The flattened rotation curves of galaxies are non-Keplerian [24], and it has been observed that a departure of the rotation curve from Newtonian gravity sets in whenever the observed acceleration falls below the following universal value, $a_M$ [4]:

$$a_M = (1.2 \pm 0.2) \times 10^{-8} \text{ cm s}^{-2}, \qquad a_M \approx \frac{1}{6}a_0 \approx \frac{1}{6}cH_0 \tag{3}$$

where $a_0$ is the observed cosmic acceleration. This discrepancy between Newtonian gravitation and the observed rotation curves can be explained by postulating that galaxies are surrounded by halos of dark matter. However, it seems difficult to understand why the dark matter distribution becomes important precisely below the above-mentioned critical acceleration (instead of beyond a critical distance from the galactic centre) and why this critical acceleration should be so close to the observed cosmic acceleration. There is also the possibility that a new fundamental force (let us call it the fifth force) becomes more significant than Newtonian gravitation whenever the acceleration $a$ falls far below the critical acceleration $a_M$. With this in mind, in 1983, Milgrom proposed that the acceleration $a$ experienced by a test body of mass $m$ in the presence of a source mass $M$ could be given by the following phenomenological law:

$$a = a_N = \frac{GM}{R^2} \quad \text{for } a \gg a_M, \qquad a = \frac{\sqrt{GMa_M}}{R} \quad \text{for } a \ll a_M \tag{4}$$

In other words, the fifth force starts to dominate over Newtonian gravitation at sub-critical accelerations. This proposal is known as Modified Newtonian Dynamics (MOND) [4]. We do not interpret it as the breakdown of Newtonian gravitation/general relativity at low accelerations, but rather as the fifth force dominating Newtonian gravity. The test body of mass $m$ universally experiences Newtonian gravity along with the fifth force due to the presence of the mass $M$. The acceleration due to both forces is independent of the mass $m$ of the test particle, but the fifth force is proportional to the square root of the source mass $M$ and falls inversely with distance ($\sim \sqrt{M}/R$) as if it were the square root of Newtonian gravitation ($\sim M/R^2$). Subsequently, we view the MOND relation $a^2 = a_N a_M$ as a consequence of the introduction of the effective distance $R_{eff}^2 = RR_H$. This latter choice makes MOND analogous to Coulomb's law and paves the way for relativistic MOND as a $U(1)$ symmetry with the square root of mass as its source.

An analogy could be made to the electroweak symmetry, which breaks down to the weak force and electrodynamics. An electron in the presence of another electron experiences both the weak force and the much stronger Coulomb force. At energy scales approaching

the electroweak scale of ∼1 Tev, the two forces have nearly equal strength and then become unified. At lower energies, the electric force dominates over the weak force, but this does not mean that the weak force law breaks down at low energies; it just means that the weak force is comparatively weaker. Analogously, the MOND force (i.e., the fifth force) dominates over Newtonian gravity (GR) at low accelerations, but this does not imply that GR is breaking down. In our work, MOND is to gravitation what electrodynamics is to the weak force. Electrodynamics (MOND) dominates over the weak force (GR) at low energies (accelerations). At high accelerations, the fifth force and GR unify (the dark electro-grav symmetry $SU(2)_R \times U(1)_{YDEM}$).

The MOND phenomenology cannot derive the interpolating function between the Newtonian regime and the deep MOND regime: that can only come from the deeper theory from which MOND originates. Thus, one introduces the unspecified interpolation function $\mu(x)$, relating the Newtonian acceleration $a_N$ to the MOND acceleration $g$ as

$$a_N = g\mu(g/a_M), \qquad \mu(x) = 1 \text{ for } x \gg 1, \quad \mu(x) = x \text{ for } x \ll 1 \tag{5}$$

If one does not wish to introduce MOND as a fifth force, it can be presented as a modified Poissonian gravity [25] by modifying the left-hand side of the Poisson equation, with $a = -\nabla\phi$.

$$\nabla.[\mu(|\nabla\phi|/a_0)\nabla\phi] = 4\pi G\rho \tag{6}$$

This modified Poisson equation can be derived from the following Lagrangian [26]:

$$\mathcal{L} = -\frac{1}{12\pi G a_0}\left((\nabla\phi)^2\right)^{3/2} + \rho\phi \tag{7}$$

Khoury notes the following: "However, as a theory of a fundamental scalar field, the non-analytic form of the kinetic term is somewhat unpalatable". For the same reason, one might be sceptical about modifying the left-hand side of the Poisson equation; doing so will make it harder to relate MOND to other fundamental interactions and to find a generalisation of GR that reduces to MOND in the non-relativistic limit at low accelerations. We prefer to derive MOND from a Poisson equation in which the left-hand side is intact as the conventional Laplacian and the right-hand side is a new source charge for a fifth force.

Nonetheless, Milgrom [25] wrote the following:

"Very interestingly, its deep-MOND limit,

$$\nabla.[(|\nabla\phi|)\nabla\phi] = 4\pi G a_0\rho \tag{8}$$

is invariant under space conformal transformations (Milgrom, 1997) [27]: Namely, besides its obvious invariance to translations and rotations, Equation (8) is invariant to dilatations, $\mathbf{r} \to \lambda\mathbf{r}$) for any constant $\lambda > 0$ [under which $\phi(\mathbf{r}) \to \phi(\mathbf{r}/\lambda)$], and to inversion about a sphere of any radius $a$, centred at any point $\mathbf{r}_0$, namely, to

$$\mathbf{r} \to \mathbf{R} + \frac{a^2}{|\mathbf{r} - \mathbf{r}_0|^2} \tag{9}$$

with $\phi(\mathbf{r}) \to \hat{\phi}[\mathbf{r}(\mathbf{R})]$, and $\rho(\mathbf{r}) \to \hat{\rho}(\mathbf{R}) = J^{-1}\rho[\mathbf{r}(\mathbf{R})]$, where $J$ is the Jacobian of the transformation (9). This ten-parameter conformal symmetry group of Equation (8) is known to be the same as the isometry (geometric symmetry) group of a 4-dimensional de Sitter spacetime, with possible deep implications, perhaps pointing to another connection of MOND with cosmology (Milgrom, 2009a) [28]."

This important fact about MOND is very encouraging for us because our proposed $U(1)_{DEM}$ symmetry is indeed the leftover unbroken symmetry from the de Sitter-like phase that precedes the dark electro-grav symmetry breaking. This correspondence with de Sitter provides justification for the use of the effective distance $R_{eff}^2 = RR_H$ because doing so enables the aforesaid invariance under dilatations.

MOND can also be presented as a modification of the law of inertia instead of as a modification of the law of gravitation:

$$a = \frac{GM}{R^2} \text{ for } a \gg a_M; \qquad \frac{a^2}{a_M} = \frac{GM}{R^2} \text{ for } a \ll a_M \tag{10}$$

In our proposal in this paper, MOND arises from a new (fifth) force obeying a modified law of inertia. Thus, both the law of gravitation and the law of inertia are modified at low accelerations.

There have been several serious attempts to develop relativistic MOND, i.e., to generalise general relativity to a modified relativistic theory of gravitation, from which MOND will emerge in the non-relativistic approximation, for accelerations $a \ll a_M$. These include the TeVeS theory developed by Bekenstein [29], which includes a vector field and a scalar field in addition to the spacetime metric. TeVeS was originally claimed to be able to explain gravitational lensing and other cosmological observations but is seriously constrained by observations in the solar system and binary stars [25]. Another prominent relativistic MOND was proposed by Skordis and Zlosnik, dubbed RMOND, which was claimed to be able to explain CMB anisotropies and the matter power spectrum [30].

Our reservation about these otherwise noteworthy relativistic generalisations is that they were expressly designed to meet the requirements of a relativistic MOND and are not easy to motivate from first principles. The vector field and scalar field introduced in TeVeS are difficult to relate to the standard model of particle physics. The quantum field theoretic constraints on such theories are also challenging. On the other hand, the $U(1)_{DEM}$ proposed by us as RelMOND is a fallout of the $E_8 \times E_8$ unification and was not designed to explain MOND. The use of $\sqrt{m}$ comes from consideration of masses of quarks and leptons of the first fermion generation. Furthermore, the $SU(2)_R \times U(1)_{YDEM}$ gauge symmetry is likely to be a renormalizable quantum field theory.

There is an extensive paper and review on MOND and its extensive applications; we do not intend to review it here. The excellent SCHOLARPEDIA article by Milgrom is up to date and reviews MOND and its applications in all its aspects [25].

However, we make mention of ongoing related research of great importance: testing the law of gravitation in GAIA DR3 wide binaries [31–33]. A large number of such binaries are known in the solar neighbourhood of the Milky Way and have orbital radii ranging from about 200 AU to 30,000 AU. The orbital acceleration crosses the critical MOND value $a_M$ for radii around 1000 AU, transiting from the Newtonian regime (relatively low radii) to the alleged MOND regime (large radii). Around 2000 AU onwards, the measured acceleration should disagree with Newtonian prediction if MOND is right. The analyses by Chae [34,35] and Hernandez [36,37] show that Newtonian gravitation is obeyed in the not-so-wide binaries but breaks down for larger separations. However, Banik et al. disagreed [38] (see Chae's critical response to Banik [35] and the responses of Lasenby, Boyle, and especially Hernandez, after the recent OSMU23 lecture by Banik [39]; see also the recent rebuttal by Hernandez and Chae [40]). Our understanding is that the conclusions of both Chae and Hernandez, that Newtonian gravitation breaks down below the critical acceleration $a_M$, are correct. It is remarkable that wide binaries have the same critical acceleration scale $a_M$ as spiral galaxies: there is no a priori reason for this to be so unless the fifth force does indeed exist and begins to dominate over gravitation below $a_M$. This anomaly in wide binaries cannot be explained by dark matter; therefore, wide binaries are the likely smoking gun that will discriminate MOND from dark matter.

We also note that not all researchers agree on the presence of a universal critical acceleration scale in galaxies. In their analysis of 193 disk galaxies from the SPARC and THINGS databases, Rodrigues et al. concluded the absence of a fundamental acceleration scale in galaxies [41], whereas in an accompanying paper, Gaugh et al. performed a Bayesian analysis1 on galaxy rotation curves from the SPARC database, finding strong evidence for a characteristic acceleration scale [42] (see also the critical analysis in [43] and the references therein). Our outlook is that these analyses are part of an ongoing

debate, and for the purpose of our present theoretical discussion, we will assume that a fundamental acceleration scale does exist on galactic scales.

Dark matter is a cornerstone assumption in contemporary cosmology, supported by evidence from galaxy rotation curves, gravitational lensing of bullet clusters, CMB fluctuations, baryon acoustic oscillations (BAOs), and the formation of large-scale structures. Therefore, it is important to assess how MOND fares vis-a-vis these aspects as an alternative to dark matter. MOND does spectacularly well with galaxy rotation curves, being able to predict the rotation curve once the baryonic mass distribution of a galaxy is known from observations. In this regard, it does better than the cold dark matter hypothesis, where a rotation curve has to be first known from observations, and then a CDM distribution is assumed so as to fit the curve of velocities. MOND also provides a natural explanation for the Tully–Fisher relation, which is a challenge for CDM. MOND does not adequately account for cluster velocity dispersions, whereas CDM certainly does better. However, it has been suggested that MOND's shortfall on cluster scales could be accounted for by the missing baryons that are currently unaccounted for in clusters.

CDM has a definite upper hand when it comes to the formation of large-scale structures, CMB anisotropies, BAOs, and gravitational lensing. Here, one is in need of a convincing relativistic MOND that generalises general relativity at low accelerations and can then be convincingly applied in cosmology. The present proposal for dark electromagnetism is a step in that direction. For a detailed recent review of the status of MOND in astrophysics and cosmology, we refer the reader to Banik and Zhao [44].

## 4. A Brief Review of Verlinde's Entropic Derivation of MOND

### 4.1. Introduction

Roughly 95% of our Universe consists of a nonbaryonic form of matter/energy exhibiting mysterious properties. This is sufficient reason to suspect that perhaps our knowledge of gravity is incomplete. General relativity might not be universally applicable (i.e., not in all regimes of parameter space), and spacetime might not be an irreducible, primary concept. Instead, our *macroscopic* notions of spacetime and gravity might emerge from an underlying *microscopic* description.

Verlinde [1,45] suggested that the observed dark energy and the phenomena usually attributed to dark matter have a common origin and can both be connected to *the emergent nature of spacetime*. The key idea is the competition between bulk degrees of freedom and surface degrees of freedom in a de Sitter universe containing matter:

(i)   When surface degrees of freedom dominate the expression for the entanglement entropy (of spacetime plus matter), we have the standard GR regime;

(ii)  When bulk degrees of freedom (in the entanglement entropy) take over, we enter the MOND regime.

This state of affairs corresponds to a *glassy* dynamics, i.e., a mechanics for the microscopic qubits of information in which two time scales are at work:

(i)   Fast, short-range dynamics that are responsible for the area law for the entanglement entropy;

(ii)  Slow, long-distance dynamics that exhibit slow relaxation, ageing, and memory effects, and are responsible for the MOND regime. This is the dark matter phase of emergent gravity.

The transition between the two occurs precisely as one crosses the cosmological horizon of de Sitter spacetime. Verlinde interpreted this as a true *phase transition* in the thermodynamic sense. The thermodynamic medium here would be a *d*-dimensional spacetime (de Sitter spacetime containing matter) exhibiting two phases:

(i)   The GR regime, corresponding to a *stiff* phase of this medium;

(ii)  The MOND regime, corresponding to an *elastic* phase.

In this picture, dark matter is *not* to be understood as being made up of some kind of particles. Rather, due to this phase transition in the fabric of spacetime itself, gravitational effects cease to be described by GR in order to exhibit MOND-like properties. GR regards spacetime as perfectly stiff; now we see that it can have elastic properties too. MOND is a consequence of the extremely small, but nonvanishing, elastic properties of spacetime. The net result is that *dark matter is an apparent phenomenon*, as its effects can be more economically understood in terms of the elastic properties of spacetime in this regime.

Altogether, Verlinde claimed that:

(i)    Spacetime emerges from the entanglement of qubits of information;
(ii)   The short-range entanglement between neighbouring bits produces an entropy scaling, as in the Bekenstein–Hawking area law;
(iii)  The long-range entanglement entropy (also called de Sitter entropy) gives rise to a volumetric law (in contrast to an area law);
(iv)   de Sitter entropy is equipartitioned between all bits;
(v)    Gravity is the force that describes the change in entanglement (i.e., in spacetime) due to matter.

*4.2. The Flattening of Galaxy Rotation Curves*

The flattening of galaxy rotation curves occurs only when the gravitational acceleration $GM/R^2$ drops below a certain acceleration scale $a_M$, i.e., whenever

$$\frac{GM}{R^2} < a_M \tag{11}$$

Here, $a_M$ is Milgrom's acceleration scale [4,46], related to the cosmic acceleration scale $a_0$, as $a_M = a_0/6$, and

$$a_0 = cH_0 \simeq 10^{-10} \; ms^{-2} \tag{12}$$

where $H_0$ is the Hubble constant.

Denoting the observed gravitational acceleration by $g_{obs}$ and the acceleration due to baryonic matter by $g_{bar}$, Milgrom proposed that $g_{obs}$ is a certain function $f$ of $g_{bar}$ such that

$$g_{obs} = f(g_{bar}) = \begin{cases} g_{bar} & \text{for} \quad g_{bar} \gg a_M \\ \sqrt{g_{bar}a_M} & \text{for} \quad g_{bar} \ll a_M \end{cases} \tag{13}$$

Equation (13) can be regarded as an equivalent restatement of the Tully–Fisher law.

Altogether, we have two extreme regimes:

(i)    When $a \gg a_M$, we have the standard Newtonian regime;
(ii)   When $a \ll a_M$, we have the MOND regime, where Newton's second law is modified.

In the intermediate regime $a \simeq a_M$, MOND makes no assumptions regarding the function $f$.

Verlinde's analysis [45] applies to de Sitter (dS) spacetime because dS is the space that best fits our Universe according to current data. Now, $d$-dimensional dS spacetime has the metric

$$ds^2 = -f(r)dt^2 + \frac{dr^2}{f(r)} + r^2 d\Omega_{d-2}^2, \qquad f(r) = 1 - \frac{r^2}{L^2}. \tag{14}$$

All computations are performed under the assumption of spherical symmetry. At $r = L$, there is a cosmological horizon that carries a finite entropy and temperature. The surface acceleration $\kappa$ is given in terms of the Hubble parameter $H_0$ and the Hubble scale $L$ as

$$\kappa = cH_0 = \frac{c^2}{L} = a_0 \tag{15}$$

Then, we can state the following:

(i)    At scales much smaller than the Hubble radius *L*, gravity is well described by general relativity (GR) because the entanglement entropy follows the Bekenstein–Hawking area law;

(ii)   At large distances, GR breaks down and MOND sets in. This corresponds to the de Sitter entropy (which follows a volumetric law) taking over.

Equivalently, we can state the following:

(i)    Gravity at accelerations greater than $a_M$ obeys GR. Spacetime in this regime, although dynamical, is regarded as *stiff*, meaning that it is *nonelastic*;

(ii)   This is opposed to gravity at accelerations below the scale $a_M$: in this MOND regime, gravity is modelled in Ref. [1] as being due to the *elastic* properties of spacetime.

In GR, the definition of mass can be problematic. Strictly speaking, the ADM mass can only be defined at spatial infinity. However, dS spacetime has a cosmological horizon and no spatial infinity. In dS spacetime, an approximate analogue of the ADM mass can be defined under some assumptions. It turns out to be given by

$$M = -\frac{1}{8\pi G} \int_{\mathcal{S}_\infty} \phi \, \partial_j n_j \, dA \tag{16}$$

where $dA = n_i \, dA_i$, $n_i$ is the outward normal vector to the surface $\mathcal{S}_\infty$ and the latter is a large-enough spherical surface enclosing the mass *M* placed around the origin. This surface $\mathcal{S}_\infty$ must be far away from the origin, so the field produced by *M* can be approximately spherically symmetric; at the same time, $\mathcal{S}_\infty$ may not be too close to the horizon.

*4.3. Entropy as a Criterion for a Phase Transition*

4.3.1. Entropy Increase When a Bit Traverses a Horizon

The addition or subtraction of *n* bits (entering the horizon or leaving it) causes an increase or decrease $\Delta S$ in the entropy of the horizon. From Verlinde's first paper [45], we have the following result for the entropy increase of a horizon, as the latter is traversed by *n* bits of information:

$$\frac{\Delta S}{n} = -k_B \frac{\Delta \phi}{2c^2}. \tag{17}$$

Here, $k_B$ is Boltzmann's constant (hereafter, $k_B = 1$) and $\Delta \phi$ is the difference in Newtonian potential between the states *before* the *n* bits traverse the horizon and *after* traversing it.

Thus, the Newtonian potential $\phi$ keeps track of the depletion of horizon entropy per bit of information traversing it.

4.3.2. Entropy of Empty dS Space

de Sitter spacetime has a certain microscopic structure, the precise form of which is unknown (and, fortunately, also irrelevant for our purposes). As a consequence, we can assign dS spacetime an entropy. For the moment, we regard dS spacetime as being *empty* or devoid of matter. The expansion of empty dS spacetime is being driven by dark energy. Verlinde computes the entropy of empty dS spacetime to be [1]

$$S_{DE}(r) = \frac{r}{L} \frac{A(r)}{4G\hbar}, \qquad A(r) = \Omega_{d-2} r^{d-2} \tag{18}$$

The subindex DE stands for *dark energy*. The above equation expresses the entropy contained within a sphere of radius *r* and surface area $A(r)$. We draw attention to the *volume* dependence on the right-hand side of (18) because of the product $rA(r)$. Happily, when evaluated at $r = L$, Equation (18) yields back the area-dependent Bekenstein–Hawking entropy. However, $S_{DE}$ scales with the volume for $r < L$.

The entropy $S_{DE}$ is carried by excitations of the qubits making up empty dS space, which lifts the negative ground-state energy to the positive value associated with the dark energy. In other words, dS entropy corresponds to the dark energy that drives the expansion of the Universe.

### 4.3.3. Entropy Reduction of dS Space due to the Addition of Matter

Our actual Universe is, of course, not empty. Applying the Bekenstein upper bound [47], Verlinde showed that the addition of a mass $M$ causes the entropy of dS space to *decrease* by the amount

$$S_M(r) = -\frac{2\pi M r}{\hbar} \tag{19}$$

because the size of the horizon is being reduced. The entanglement between the two sides of the horizon diminishes due to the addition of this mass, hence the negative entropy.

### 4.3.4. The Missing Mass Problem in Entropic Terms

We return to Equation (11), which we would like to re-express in entropic terms. Consider a spherical region with boundary area $A(r) = 4\pi r^2$ containing a total mass $M$. Then, the gravitational phenomena attributed to dark matter occur only when the *area density* $\Sigma(r)$ *of mass* falls below a universal value determined by $a_M$:

$$\Sigma(r) = \frac{M}{A(r)} < \frac{a_M}{8\pi G} \tag{20}$$

We have replaced condition (11), expressed in terms of accelerations, with condition (20), expressed in terms of the surface density of mass. Next, we recast the same condition in terms of entropies. For this, we rewrite (20) more suggestively as

$$\frac{2\pi M}{\hbar a_M} < \frac{A(r)}{4G\hbar}, \tag{21}$$

We multiply through with $r/L$ and use (15) to obtain

$$\frac{2\pi M r}{\hbar} < \frac{r}{L}\frac{A(r)}{4G\hbar}. \tag{22}$$

Finally, using (18) and (19), we have

$$|S_M(r)| < S_{DE}(r). \tag{23}$$

To summarise, the gravitational phenomena commonly attributed to dark matter occur whenever the inequality (23) holds. The bulk entropy $S_{DE}$ scales with the volume, whereas the matter entropy $S_M$ scales linearly with $r$. The observations on galaxy rotation curves tell us that the nature of gravity changes, depending on whether the matter added to dS space removes all or just a fraction of the entropy $S_{DE}$ of dS space.

Therefore, we have two regimes:

(i) The regime when $S_M(r) < S_{DE}(r)$, which corresponds to low surface mass density $\Sigma(r)$ and low gravitational acceleration. This is the MOND or sub-Newtonian or *dark matter* regime;

(ii) The regime when $S_M(r) > S_{DE}(r)$, which describes Newtonian gravity.

Verlinde's goal was to explain why the laws of emergent gravity differ from those of general relativity (GR) precisely when the inequality (20) (equivalently, (23)) holds. His conclusions were as follows:

(i) At scales much smaller than the Hubble radius, gravity is well described by GR because the entanglement entropy is still dominated by the area law of the vacuum; this is identified as the *stiff* phase of spacetime;

(ii) At larger distances and/or longer time scales, the bulk dS entropy leads to modifications of the above laws. Precisely when the surface mass density falls below the value (20), the reaction force due to the thermal contribution takes over from the usual gravity governed by the area law. This is identified as the *elastic* phase of spacetime, in which MOND gravity takes over.

### 4.3.5. Newtonian Gravity in Terms of Surface Densities

Motivated by the previous arguments, next, we rewrite the familiar laws of Newtonian gravity in terms of a *surface mass density* vector $\boldsymbol{\Sigma}$.

Given a Newtonian potential $\phi$ and the corresponding acceleration $g_i = -\partial_i \phi$, we define

$$\Sigma_i = \frac{d-2}{d-3} \frac{g_i}{8\pi G} \tag{24}$$

where $g_i$ is the usual gravitational acceleration vector with some convenient normalisation. The latter is chosen so that the differential expression of Gauss's law in $d$-dimensional dS spacetime now reads as

$$\nabla \cdot \boldsymbol{\Sigma} = \rho \tag{25}$$

where $\boldsymbol{\Sigma}$ qualifies as a surface mass density, which follows from the equivalent integral expression of the Gauss law

$$\int_{\mathcal{S}} \boldsymbol{\Sigma} \cdot d\mathbf{A} = M \tag{26}$$

where $M$ is the total mass enclosed by the surface $\mathcal{S}$. Finally, the gravitational self-energy $U_{\text{grav}}$ of a mass distribution can also be expressed in terms of $\boldsymbol{\Sigma}$:

$$U_{\text{grav}} = \frac{1}{2} \int dV \, g_i \Sigma_i \tag{27}$$

This rewriting of Newtonian gravity in terms of surface densities facilitates its interpretation in terms of elasticity theory.

### 4.4. The Elastic Phase of Emergent Gravity

4.4.1. Elastic Moduli in Terms of Gravitational Parameters

Verlinde next proved that the ADM-like definition of mass (16) can be naturally translated into an expression for the strain tensor. Given the Newtonian potential $\phi$, the corresponding elastic displacement field $u_i$ is postulated to be

$$u_i = \frac{\phi}{a_0} n_i \tag{28}$$

where $n_i$ is the outward unit normal to a surface $\mathcal{S}_\infty$. The latter encloses a mass given by

$$M = \frac{a_0}{8\pi G} \int_{\mathcal{S}_\infty} \left( n_j \varepsilon_{ij} - n_i \varepsilon_{jj} \right) dA_i \tag{29}$$

where $\varepsilon_{ij}$ is the strain tensor for the displacement field $u_i$. Multiplying both sides of (29) by the acceleration scale $a_0$, we obtain a force

$$M a_0 = \int_{\mathcal{S}_\infty} \sigma_{ij} n_j dA_i \tag{30}$$

where we have identified the stress tensor of the dark matter elastic medium to be

$$\sigma_{ij} = \frac{a_0^2}{8\pi G} \left( \varepsilon_{ij} - \varepsilon_{kk} \delta_{ij} \right) \tag{31}$$

This yields *the elastic moduli of the dark matter medium*:

$$\mu = \frac{a_0^2}{16\pi G}, \qquad \lambda = -\frac{a_0^2}{8\pi G} \tag{32}$$

4.4.2. A Derivation of the Tully–Fisher Relation

Dark matter causes a gravitational pull, an acceleration $g_D$, which scales with $\sqrt{M_D}$, the square root of the dark mass $M_D$. This is in contrast to baryonic matter, whose acceleration $g_B$ scales with the baryonic mass $M_B$. Verlinde found that in $d$-dimensional dS spacetime, one has the following analogue of (13):

$$g_D^2 = g_B \, a_M, \qquad a_M = \frac{d-3}{(d-2)(d-1)} a_0 \tag{33}$$

When $d = 4$, Equation (33) is equivalent to the Tully–Fisher relation (13). Then, $a_M = a_0/6$, which is the acceleration scale appearing in Milgrom's phenomenological fitting Formula (13).

Equation (33) is *a theoretical* derivation of the *phenomenological* Tully–Fisher law, Equation (13). This derivation from first principles can be seen as one of the main achievements of Verlinde's paper.

4.4.3. Apparent Dark Matter in Terms of Baryonic Matter

Using the previous definitions of elastic and gravitational quantities, Verlinde derived an expression for the density of apparent dark matter as a function of baryonic matter, namely $\Sigma_D$ as a function of the *baryonic* Newtonian potential $\phi_B$:

$$\left( \frac{8\pi G}{a_0} \Sigma_D \right)^2 = \left( \frac{d-2}{d-1} \right) \frac{1}{a_0} \partial_i (\phi_B \, n_i) \tag{34}$$

In the spherically symmetric case and when $d = 4$, the above can be integrated within a sphere of radius $R$ to yield

$$\int_0^R \frac{G M_D^2(r)}{r^2} \, dr = \frac{1}{6} M_B(R) a_0 R \tag{35}$$

where

$$M(R) = \int_0^R \rho(r) A(r) \, dr \tag{36}$$

is the total mass inside the radius $R$. Equations (34) and (35) describe *the amount of apparent dark matter in terms of the amount of baryonic matter*. As such, they allow us to make direct comparisons with observations. In Ref. [48], it was claimed that this agreement is good.

**5. Dark Electromagnetism as the Origin of Relativistic MOND**

We demonstrate that MOND can be written as Coulomb's law analogous to Maxwell's electrodynamics by using an effective distance. Energy conservation, along with a modified inertia law, can then be used to show that, when written as Coulomb's law, MOND mimics cold dark matter, including on cosmological scales. Furthermore, in the deep MOND regime, this formulation is the non-relativistic limit of the $U(1)_{DEM}$ gauge theory.

We have in the deep MOND regime that the acceleration $a$ of a test particle in the field of a mass $M$ is given by

$$a = \sqrt{G a_M} \, \frac{\sqrt{M}}{R} = \sqrt{L_P c^2 a_M} \, \frac{\sqrt{M/m_{Pl}}}{R} = c^2 \sqrt{\frac{L_P}{L_M}} \, \frac{\sqrt{M/m_{Pl}}}{R} \tag{37}$$

where $a_M$ is Milgrom's acceleration constant and $L_M = c^2/a_M$ is the MOND radius.

We will assume that the MOND force $F$ on the test particle of mass $m$ can be obtained by multiplying the acceleration by $\sqrt{m m_{Pl}}$. We write the force in terms of dimensionless masses to try to make it look more and more like electrodynamics:

$$F = \sqrt{m m_{Pl}} \, a = m_{Pl} c^2 \sqrt{\frac{L_P}{L_M}} \, \frac{\sqrt{M/m_{Pl}} \, \sqrt{m/m_{Pl}}}{R} \tag{38}$$

Assuming that we live in a de Sitter universe, we multiply and divide by the Hubble radius $R_H = cH_0^{-1}$, which is also the de Sitter horizon, and we introduce the effective distance $R_{eff}$ given by $R_{eff}^2 \equiv RR_H$.

$$F = \sqrt{mm_{Pl}}\, a = m_{Pl}c^2 R_H \sqrt{\frac{L_P}{L_M}} \frac{\sqrt{M/m_{Pl}}\,\sqrt{m/m_{Pl}}}{R_{eff}^2} \tag{39}$$

Now, this looks like Coulomb's law in terms of the effective distance $R_{eff}$. If a spatial point **x** is at a distance $|\mathbf{x}|$ from the observer, it has to be stretched by a factor $R_H$. We can discuss the covariance of this procedure, but in a Robertson–Walker universe with cosmic time, this procedure seems well defined.

We assume that the Milgrom constant $a_M$ is $\eta$ times the cosmic acceleration $a_0 = cH_0$ and also that $a_0 = \beta a_{Pl}(L_P/R_H)$ is the scaling down of the Planck acceleration due to the de Sitter expansion. Thus, $L_M = c^2/a_M = c^2/\eta a_0 = c^2 R_H/\eta\beta a_{Pl}L_P$. We can hence write the force as

$$F = A \frac{\sqrt{M/m_{Pl}}\,\sqrt{m/m_{Pl}}}{R_{eff}^2} \tag{40}$$

where

$$A = m_{Pl}c\sqrt{R_H}\sqrt{L_P^2 a_{Pl}\eta\beta} = \hbar c\sqrt{\eta\beta R_H/L_P} \longrightarrow A = \frac{3}{2}\hbar c\sqrt{\eta\beta R_H/L_P} \tag{41}$$

The factor of $3/2$ is deliberately introduced to achieve consistency with Verlinde's result and consistently derive the famous factor of $1/6$, which relates Milgrom's constant to the cosmic acceleration. We will take (40) as the defining force law of the $U(1)_{DEM}$ interaction, with $A$ as defined in (41), including the factor of $3/2$. MOND is to be derived from this force law, even though initially, we started from MOND to motivate this Coulomb-like force law.

Below, we consider generalising this to a fully relativistic theory for the square-root mass current. The theory can be expected to be derivable from an action principle, similar to Maxwell electrodynamics. For now, let us continue with the spherically symmetric Coulomb case.

This force law has an interesting parallel with, and an important difference from, Maxwell's electrodynamics. We can write Coulomb's law as $F = \hbar c(e^2/\hbar c)/R^2$. Here, the charge is expressed in dimensionless units, so multiplication by $\hbar c$ appears, similar to the above gravity case. However, the gravitational coupling is scaled by a factor dependent on the epoch via the Hubble radius (with the understanding that $R_H = c^2/a_0$, and any epoch dependence would come from the in-principle-allowed time variation of the cosmic acceleration). Gravity uses the effective distance, which is like a scaling of the actual distance.

The introduction of a characteristic acceleration ($a_M$) related to the Hubble constant implies variability over time, suggesting that galaxies at different redshifts would exhibit distinct rotation curves. This aspect has been carefully discussed, for instance, in the MOND review by Milgrom [25] (see the subsection on 'The Significance of the MOND Acceleration Constant'; see also [49], where high-redshift rotation curves are discussed in the context of MOND). Milgrom concluded that there are strong observational constraints on the variation of $a_M$ with cosmic time, and a value of $4a_M$ at $z \sim 2$ was essentially ruled out, hence excluding a dependence such as $a \propto (1+z)^{3/2}$.

The force law can be derived from a potential $\phi$ via $F = d\phi/dR_{eff}$ so that

$$\phi = -A \frac{\sqrt{M/m_{Pl}}\,\sqrt{m/m_{Pl}}}{R_{eff}} \tag{42}$$

We would now like to note the energy conservation equation in the deep MOND regime given this potential, and from this equation derive Verlinde's central equation in his

paper [1], Equation (7.40). The energy conservation equation is obtained by starting from the equation of motion for the test mass $m$ at $R$ with a velocity of $v = dR/dt$.

$$\sqrt{m}\sqrt{m_{Pl}}\dot{v} = -\frac{d\phi}{dR_{eff}} = -\frac{dR}{dR_{eff}}\frac{d\phi}{dR} \tag{43}$$

The left-hand side of this equation is a modified inertia law and, in fact, is such that the MOND acceleration is independent of the square-root mass of the test particle. Thus, we still have the equivalence principle, but this time arising from the cancellation of square-root mass when the dark charge is identified with the inertial square-root mass.

Multiplying both sides by $v$ and noting that $dR_{eff}/dR = R_H/2R_{eff}$, we can write

$$\sqrt{m_{Pl}}\sqrt{m}\,\frac{1}{2}\frac{d}{dt}[v^2] + \frac{2R_{eff}}{R_H}\frac{d}{dt}\phi_{eff} = 0 \tag{44}$$

If we make the crucial assumption that the time dependence of $2R_{eff}/R_H$ can be ignored, this equation can be integrated to obtain the following expression for conserved energy after substituting the form of the potential:

$$-\frac{2E}{\sqrt{m}\sqrt{m_{Pl}}}\frac{1}{R^2} + \frac{\dot{R}^2}{R^2} = \frac{6\sqrt{Ga_0\eta}\sqrt{M}}{R^2} \tag{45}$$

As done in the Newtonian derivation of the Friedmann equation (converting force law into energy conservation), we equate the right-hand side term to the source term of the Einstein equations, as if sourced by an apparent dark matter distribution $M_D(R)$ (Verlinde's notation):

$$\frac{6\sqrt{Ga_0c\eta}\sqrt{M}}{R^2} = 8\pi G\rho_D = 8\pi G\frac{M_D}{4\pi R^3/3} = \frac{6GM_D}{R^3} \tag{46}$$

assuming a constant density and a uniform apparent dark matter distribution. Squaring both sides gives

$$\frac{GM_D^2}{R} = \eta a_0 MR \tag{47}$$

which is consistent with Verlinde's equation in [1], Equation (7.40), if we assume $\eta = 1/6$. From here, following Verlinde, the MOND law can be easily derived.

It seems interesting that we yield the same result for apparent dark matter as Verlinde. This can be considered support for the proposed $U(1)_{DEM}$ symmetry. Furthermore, the introduction of the effective distance can be interpreted as a stretching of the distance $R$ to the larger distance $R_{eff}$ and reminds us of an elastic medium. We should explore how to relate our effective distance to Verlinde's elasticity approach to MOND, as the two might be related to each other. Note that the amount of apparent dark matter $M_D$ is proportional to the square root of the actual matter $M$. We hope to derive these results from first principles in future work.

Now, we can also try to prove that the total amount of apparent dark matter is about five times that of ordinary matter. Verlinde's Equation (7.40) is

$$\int_0^R \frac{GM_D(r')^2}{R'^2} = \frac{1}{6}a_0 M(R)R \tag{48}$$

Assuming a uniform density $\rho_D$, one can integrate the left-hand side after expressing mass in terms of density, to obtain

$$\frac{GM_D^2}{5R} = \frac{1}{6}a_0 M_D(R)R \longrightarrow M_D = \sqrt{\frac{5}{6G}}\sqrt{a_0 MR} = \sqrt{\frac{5}{3}}M \tag{49}$$

The last equality follows by considering the entire Universe and writing the mass $M$ in terms of density, assuming critical density $\rho = 3H_0^2/8\pi G$, which gives $H_0 = 1/2GM$. For $R$, we have assumed that the value of the Hubble radius is $R_H = H_0^{-1}$, which is also the de Sitter horizon.

This is the contribution to apparent dark matter from the Coulomb part of the potential energy. If we assume that each of the three vector components also contributes equally, we deduce that the total apparent dark matter is $4 \times \sqrt{5/3} = 5.16$ times ordinary matter. This agrees well with the standard LCDM model, according to which the dark matter to ordinary matter ratio is about 5.3. The assumption that the vector components contribute equally, as the Coulomb part, is reasonable because these considerations are being applied on the scale of the entire Universe, including at high redshifts. Therefore, relativistic motions must be taken into account, and as a result, the 'magnetic part' of the four-potential is expected to be as significant as the Coulomb part.

A very important point is that only particles with non-zero rest mass take part in dark electromagnetism, just as only particles with non-zero electric charge take part in electromagnetism. Hence, there is no $U(1)_{DEM}$ interaction between photons and baryonic matter. From this point of view, the apparent dark matter derived above perfectly mimics dark matter. It will produce an additional gravitation-like attraction but will not have any impact on the CMB anisotropy produced by baryons interacting with electromagnetic radiation on the last scattering surface. We can, as usual, study the growth of linear density perturbations by working with apparent density fluctuations in apparent dark matter.

Furthermore, the potential energy of the dark electromagnetic field serves as a source on the right-hand side of the Einstein equations, similar to cold dark matter. Therefore, insofar as causing gravitational lensing is concerned, the DEM field mimics cold dark matter.

The non-relativistic limit of dark electromagnetism (dark equivalent of Coulomb's law) proposed above is the limit of relativistic dark electromagnetism, modelled after Maxwell's electrodynamics.

We propose the following action principle for dark electromagnetism and general relativity, entirely modelled after the action for Maxwell's electrodynamics coupled to sources in curved spacetime. The electromagnetic field is replaced with the DEM field.

$$
\begin{aligned}
S = \frac{c^3}{16\pi G} \int d^4x \sqrt{-g}\, [R - 2\Lambda] + \int d^4x \sqrt{-g}\, \mathcal{L}_{matter} \\
- \frac{1}{16c} \int d^4x \sqrt{-g}\, F_{ij}F^{ij} + A \int d^4x \sqrt{-g} D_i J^i
\end{aligned}
\tag{50}
$$

The last term couples the dark electromagnetic potential $D_i$ to the current density $J^i$ of square-root mass, obtained by multiplying the latter by four velocities. The coupling constant $A$ was defined earlier in Equation (41), wherein $R_H$ is to be understood as $R_H = c^2/a_0$. The source for gravity is the energy-momentum tensor of mass and the energy-momentum tensor of the dark field. The dark current is given by

$$
J^i = \sum_a \frac{\sqrt{m/m_{pl}}\, c}{\sqrt{-g}}\, \delta(\mathbf{y} - \mathbf{y}_a)\, \frac{dx^i}{dx^0}
\tag{51}
$$

Here, the spatial distance $\mathbf{y}$ is the effective distance, i.e., $|\mathbf{y}|^2 = R_H|\mathbf{x}|$, whereas time $t = x^0/c$ corresponds to the cosmic time used in the Robertson–Walker metric and Friedmann equations. The dark potential is also a function of the effective spatial distance $\mathbf{y}$, not of $\mathbf{x}$. Thus, if we define $y^i = (t, \mathbf{x})$, then $F_{ij} = \partial_{yi}D_j(y) - \partial_{yj}D_i(y)$. The interaction term of the dark current does not contribute to the energy-momentum tensor that appears on the right-hand side of the Einstein equations because the $\sqrt{-g}$ in the denominator of the expression for current density cancels the $\sqrt{-g}$ in the numerator in the expression for interaction action (last term in the above action). This is the same as in Maxwell's electrodynamics, but in the present case of dark electromagnetism, it has profound significance. Specifically,

the source term for GR (being the energy-momentum tensor proportional to mass *m*) is distinct from the source term for dark electromagnetism, being the current density of square-root mass. Two masses $m_1$ and $m_2$ interact, both via GR and DEM, with one interaction dominating over the other depending on the magnitude of the acceleration. Furthermore, the introduction of the effective distance in DEM and the specific use of cosmic time breaks Lorentz invariance. DEM as relativistic MOND adopts a specific reference frame, which we define as the rest frame of the CMB.

The second-last term is the action for the DEM field, which is derived from its field tensor and also interacts with gravitation. Its energy-momentum tensor contributes as a source in the Einstein field equations. By varying the action with respect to the metric we obtain Einstein's field equations sourced by dust and the DEM field. Varying it with respect to the DEM field yields Maxwell-like equations that couple the DEM field to the current density of square-root mass. Finally, varying it with respect to particle position yields the geodesic equation of motion, which now also includes the effect of the DEM field as an external non-gravitational force.

More explicitly, variation of action yields the Einstein equations

$$R_{\mu\nu} - \frac{1}{2} g_{\mu\nu} R + \Lambda g_{ik} = \frac{8\pi G}{c^4} \left( T_{\mu\nu}(matter) + T_{\mu\nu}(DEM) \right) \tag{52}$$

and Maxwell-like equations, sourced by the current density of square-root mass, all written as functions of the effective spatial distance and cosmic time:

$$F^{ik}_{:k} = -\frac{4\pi}{c} A j^i \tag{53}$$

The solution to this equation determines the DEM field, which then enters the Einstein equations as a source, as an alternative to dark matter. This source term is the potential energy of the DEM field, and it represents the enhanced gravitational interaction amongst baryons without the need to invoke dark matter to provide the sought-after additional gravitational effects. The right-hand side of Equation (45) is an illustration of this claim.

The geodesic equation also includes an external force (so the motion becomes non-geodesic), with the Maxwell-like force being proportional to the square-root mass (analogous to the electric charge) and a function of the effective spatial distance

$$mc \frac{Du^i}{ds} = mc \left( \frac{du^i}{ds} + \Gamma^i_{kl} u^k u^l \right) = \frac{\sqrt{m/m_{Pl}}}{c} A F^{ik} u_k \tag{54}$$

As and when the effects of DEM become insignificant, Lorentz invariance and GR are recovered, as expected. These field equations reduce (in the Newtonian approximation and the homogeneous isotropic cosmological approximation) to the analysis in Section 5.

The treatment of the exact field equations is left for future work. If such an analysis can be performed, it might even yield the sought-after interpolating function that mediates between Newtonian gravitation and MOND.

Milgrom [28] wrote that "...one may conjecture that the MOND-cosmology connection is such that local gravitational physics would take exactly the deep-MOND form in an exact de Sitter universe. This is based on the equality of the symmetry groups of $dS^4$ and of the MOND limit of the Bekenstein–Milgrom formulation [50] both groups being $SO(4,1)$. The fact that today we see locally a departure from the exact MOND-limit physics, i.e., that the interpolating functions have the form they have and that $a_0$ is finite and serves as a transition acceleration, stems from the departure of our actual spacetime from exact $dS^4$ geometry: The broken symmetry of our space-time is thus echoed in the broken symmetry of local physics". Our proposal, suggesting that $U(1)_{DEM}$ is the remnant unbroken symmetry after the breaking of $SU(2)_R \times U(1)_{YDEM}$, aligns entirely with Milgrom's conjecture.

A few further remarks about the proposed action principle in Equation (50) are needed. This form of the action is assumed to come into play after the electroweak symmetry breaking around a TeV scale, that is, the epoch at which classical spacetime, general relativity, and dark electromagnetism (i.e., relativistic MOND) emerge. This emergence is expected to yield the same physical results as standard Big Bang cosmology, with cold dark matter exchanged for dark electromagnetism. Prior to this emergence, cosmology is governed by the unified $E_8 \times E_8$ theory above the TeV scale. Since cosmological data are not yet available at such high-energy scales, there is no contradiction between the octonionic theory and the cosmology of the very early Universe.

## 6. Derivation of Verlinde's Entropic Criterion from Dark Electromagnetism

Consider the epoch of left–right symmetry breaking, where the $SU(2)_R \times U(1)_{YDEM}$ symmetry is also broken. The de Sitter expansion (as in the octonionic theory) ends with the formation of compact objects. The $U(1)$ symmetry remains unbroken, like in the electroweak sector, and becomes the $U(1)_{DEM}$ symmetry, which we are currently examining. $U(1)_{DEM}$, like MOND, is a scale-invariant theory and carries the memory of the de Sitter phase. GR arises as a result of symmetry breaking. Consider a black hole arising from spontaneous localisation, which, in fact, is how the de Sitter expansion ends. As Verlinde showed [1], the formation of a localised compact object reduces the de Sitter entropy. The criterion for MOND to be dominant is that this reduction in entropy (which is area-proportional) is less than the volume entropy of de Sitter in the volume occupied by the compact object. This is equivalent to saying that the memory of de Sitter is retained under these conditions and that $U(1)_{DEM}$ dominates over GR.

We can try to derive Verlinde's entropy criterion by starting from our $U(1)_{DEM}$ theory. Let us start by asking the following question: What is the temperature of a black hole whose radius is such that its surface gravity is less than the critical MOND acceleration? Assuming that the radius $R$ of the black hole is given as in GR, hence $R = 2GM$, the acceleration on the surface is, assuming that the effective radius of the black hole is $R_{eff} = \sqrt{RR_H} = \sqrt{2GMR_H}$,

$$a = \sqrt{GMa_M}/R \longrightarrow a = \sqrt{GMa_M}/R_{eff} = \sqrt{GMa_M}/\sqrt{2GMR_H} \sim a_0 \qquad (55)$$

where we have neglected the numerical coefficient for now. The interesting point is that this acceleration is independent of the mass of the black hole. If we attribute a temperature to the black hole, proportional to the surface gravity, the temperature is $a_0$, just as for the de Sitter horizon, and independent of the mass of the black hole. This is an example of de Sitter memory being retained in the $U(1)_{DEM}$-dominated deep MOND regime.

The entropy of the black hole is given by

$$dQ = TdS \rightarrow dS = dQ/T = dM/a_0 \rightarrow S \sim M/a_0 \qquad (56)$$

which is consistent with Verlinde's result. In the deep MOND regime, this entropy is less than the de Sitter volume entropy, as a direct consequence of our $U(1)_{DEM}$ theory.

## 7. Coupling Constants in the Dark Electro-Grav Theory

For the electroweak sector $SU(2)_L \times U(1)_Y$, the derived fundamental constants are the low-energy fine structure constant $\alpha_{fsc} \equiv e^2/\hbar c$ and the weak mixing angle (Weinberg angle) $\theta_W$, with the latter being the solution of the trigonometric Equation (56) in our paper [5]. The fine structure constant is constructed from the parameters $\alpha$ and $L$ appearing in the Lagrangian of the theory, as depicted, e.g., in Equation (6) in the aforementioned paper. The constants of the electroweak sector can be expressed in terms of the fine structure constant and the weak mixing angle, along with the value of the Higgs mass $m_H$, whose value is predicted from cosmological downscaling (caused by the de Sitter-like inflationary expansion) from the original Planck scale value of the Higgs mass. It is significant that the standard model Higgs comes from the right sector in the left–right symmetric model

(whereas the standard model forces arise from the left sector). The second Higgs, associated with the left sector, is a newly predicted Higgs that is electrically charged.

Thus, the weak isospin $g$ (i.e., the $SU(2)_L$ coupling) is given by $g = e / \sin \theta_W$, and the weak hypercharge $g'$ (the $U(1)_Y$ coupling) is given by $g' = e / \cos \theta_W$. The Higgs mass is estimated as follows. When the mass ratios are computed in the octonionic theory, we assume that the lightest of the masses, i.e., the electron mass, is one in Planck units. Likewise, the charge of the down-quark, which is the smallest electric charge, is set to one while determining the fine structure constant. Hence, the Higgs mass is initially about $3 \times 10^5 m_P \sim 10^{24}$ GeV because the Higgs is a composite of standard model fermions and is expected to obtain maximum contribution from the up-quark, which, at about 173 GeV, is about $3 \times 10^5$ heavier than the electron. An inflation by a factor of $10^{61}$ scales this mass down by a factor of $10^{61/3}$ to the value of about $10^3$ GeV. This sets the weak coupling Fermi constant $G_F^0 \sim 1/v^2$ (where $v \sim 246$ GeV is the Higgs VEV) to about $10^{-6}$ GeV$^{-2}$, whereas the experimentally measured value for the Fermi constant is about $10^{-5}$ GeV$^{-2}$.

This derivation of the reduced coupling constant $G_F^0 = G_F / (\hbar c^3) \sim g^2 / M_W^2 c^4$ enables us to obtain a reasonable estimate of the W boson mass from first principles. We also observe that the Fermi constant has dimensions of length squared (same as $G_N$) and can be written as $G_F^0 \sim g^2 (m_P / m_W)^2 m_P^{-2} \sim g^2 (m_P / m_W)^2 G_N / (\hbar c)$. The scaling down of the W mass from its Planck scale value is responsible for the weak force becoming so much stronger than gravitation. In this theory, $G_N$ remains unchanged with the epoch.

Knowing $m_W$, the mass of the Z boson is determined conventionally by the relation $m_Z = m_W / \cos \theta_W$. This way, we have a handle on the fundamental constants and parameters of the electroweak sector, including the Higgs mass, Fermi constant, fine structure constant, weak mixing angle, masses of weak bosons, weak isospin, and hypercharge. To understand why there are sixty-one orders of magnitude of inflation, which ends at the electroweak scale, please see [20]. The same result is also supported by the idea that the electroweak symmetry is broken below a critical acceleration (see Section 8 below).

Let us now discuss the coupling constants and parameters of the right-handed dark electro-grav (DEM) sector, $SU(2)_R \times U(1)_{YDEM}$, staying as close as possible to the above discussion for the electroweak sector. The DEM symmetry is broken along with the electroweak symmetry. It can also be shown using the electric charge operator, i.e., the number operator that is associated with the $U(1)_{em}$ symmetry, that $W^+$ and $W^-$ have electric charges of $+1$ and $-1$, respectively, and that $Z^0$ is electrically neutral. The corresponding situation for the $SU(2)_R \times U(1)_{YDEM}$ symmetry is interesting because here, the number operator associated with $U(1)_{DEM}$ defines the square root of mass (in Planck mass units); it does not define the electric charge. Consequently, $W_R^+$ and $W_R^-$ have square root masses of $+1$ and $-1$, respectively; hence, their range of interaction is limited to the Planck length. They will also have an extremely tiny electric charge, some seventeen orders of magnitude smaller than the charge of the electron (analogous to the W mass being so small on the Planck scale). Conversely, the $U(1)_{YDEM}$ boson (and the dark photon it transforms to) will have zero mass and zero electric charge. $Z_R^0$ will be massless and will have an extremely tiny electric charge (like the $W_R$ bosons). It is possible that emergent gravitation is mediated at the quantum level by the $Z_R^0$ and the dark photon, effectively replacing the role of the spin-2 graviton in this theory.

The place of the fine structure constant is assumed by the mass of the electron. The Weinberg angle satisfies the same equation and hence has the same value as in the electroweak case. Thus, the right-sector analogues of the couplings $g$ and $g'$ can be obtained. GR is the result of the breaking of the $SU(2)_R$ symmetry (i.e., the quantum-to-classical transition). The remaining unbroken symmetry is dark electromagnetism $U(1)_{DEM}$, which is the proposed origin of relativistic MOND. The cosmological origin of MOND is briefly discussed in [20].

During the de Sitter-like inflationary phase, $E_8 \times E_8$ symmetry is operational and includes, as a subset, the unbroken electroweak symmetry $SU(2)_L \times U(1)_Y$, as well as the dark electro-grav symmetry $SU(2)_R \times U(1)_{YDEM}$. Below the critical acceleration, these

symmetries are broken, giving rise to the emergence of classical spacetime (precipitated by the localisation of fermions). Near compact objects, the gravitationally induced acceleration (GR/Newton) is higher than the critical acceleration and GR dominates. In the far zone, the acceleration is below the critical acceleration: this is the deep MOND regime where the unbroken symmetry $U(1)_{DEM}$ of dark electromagnetism dominates. This zone acts as a buffer between the de Sitter horizon and the GR zone, and it was also identified in Verlinde's work using his entropy considerations.

All left-handed particles take part in the weak force, and all electrically charged particles take part in electromagnetism. Analogously, all right-handed particles take part in the $SU(2)_R$ interaction, whereas all particles with non-zero square-root mass take part in dark electromagnetism.

## 8. Discussion

We somehow tend to think that $R$ is the genuine distance and that the effective distance $R_{eff}$ is introduced by brute force. This need not be true, and the actual situation can be the other way around. Let us rename the effective distance $R_{eff}$ as the true distance $R_{true}$. We do this for the following reason. In our approach, the Universe starts out as a de Sitter-like universe, and the formation of structures such as black holes (GR-dominated near BHs, MOND-dominated farther out) ends the de Sitter phase. Let $R_{true}$ be the physical distance of some point with respect to the observer. We propose that as a result of spontaneous localisation, which causes a classical structure such as a black hole to form, the distance $R_{true}$ shrinks to $R$ in the same ratio that the Hubble radius (event horizon distance) bears to $R_{true}$. Therefore,

$$\frac{R}{R_{true}} = \frac{R_{true}}{R_H} \tag{57}$$

This provides some physical basis, in terms of initial conditions, for using the effective distance.

### 8.1. Critical Acceleration

It has been demonstrated by previous researchers that if an inertial observer observes a spontaneously broken symmetry, a Rindler observer concludes that the symmetry is not broken, provided the acceleration is above a certain critical value (see, e.g., [51,52]). Padmanabhan was one of the researchers who demonstrated this result. Indeed, Section 7 and, in particular, Equation (7.15) in the work of Padmanabhan [53] demonstrate this effect.

The 2017 paper [54] demonstrated the critical acceleration for the electroweak case. This result appears significant for what we are doing with dark electromagnetism arising from the breaking of $SU(2)_R \times U(1)_{YDEM}$ in the early universe. It helps us understand that classicality and GR emerge as a result of the acceleration of the universe falling below the critical value. This critical value happens to be the same as the current acceleration of the universe.

These results could have important implications in early Universe cosmology. In particular, it could be that the electroweak symmetry breaks when the acceleration of a quasi-de Sitter expanding universe falls below a critical value, assuming that the inflation-like phase ends at the electroweak scale (see the critical analysis in this regard by Unruh and Weiss [55] and by Hill [56]).

In our research we are investigating whether Milgrom's MOND arises as the result of the breaking of an $SU(2)_R \times U(1)_{YDEM}$ symmetry, which is the right-handed counterpart of the electroweak symmetry. After spontaneous symmetry breaking, the $U(1)$ becomes $U(1)_{DEM}$, which is dark electromagnetism. We are examining whether this fifth force is an alternative to dark matter and provides the sought-after theoretical basis of MOND. The critical acceleration result could be relevant in establishing the SSB criterion.

### 8.2. Limiting Values

Consider the quantity $\frac{\sqrt{M}}{R_{eff}^3}$, which can be expanded around a spatial point as $\frac{\sqrt{\rho_0 R^3}}{R^{3/2} R_H^{3/2}}$, resulting in the finite limit $\sqrt{\rho_0 / R_H^3}$. This reinforces the use of the effective distance. In a non-spherical situation, the effective distance between two spatial points (with coordinate difference $\mathbf{x} - \mathbf{x}'$) is defined by new coordinates $\mathbf{y} - \mathbf{y}'$ such that $|\mathbf{y} - \mathbf{y}'|_{eff}^2 = R_H |\mathbf{x} - \mathbf{x}'|$.

### 8.3. Advantages of Considering Dark Electromagnetism

Here, we summarise some of the key advantages of $U(1)_{DEM}$ symmetry:

1.  It arises from the first principles of $E_8 \times E_8$ theory.
2.  It is a relativistic gauge theory.
3.  It is plausible that the unbroken $SU(2)_R \times U(1)_{YDEM}$ symmetry is renormalisable and is the correct theory of quantum gravity.
4.  $U(1)_{DEM}$ is sourced by the square root of mass, which is desired by MOND.
5.  Only particles with non-zero rest mass take part in $U(1)_{DEM}$. A photon does not interact with matter through $U(1)_{DEM}$. Therefore, the additional force created by the baryon–$U(1)_{DEM}$ interaction is the perfect dark matter mimicker. It will explain CMB anisotropies for the same reason that dark matter explains CMB anisotropies. It will also mimic dark matter vis-a-vis gravitational lensing.
6.  There is a natural connection with de Sitter because $U(1)_{DEM}$ is the leftover unbroken symmetry from de Sitter.
7.  We are able to derive Verlinde's results for apparent dark matter and the entropy criterion for MOND.
8.  Previous researchers have demonstrated that the electroweak symmetry is broken below a certain critical acceleration and restored above it. The same result can be expected to hold for its right-handed counterpart, this being our GR-DEM theory. Analogous to electroweak, we could call it dark electro-grav.

The dark photon—the massless gauge boson that mediates quantised DEM—can be thought of as dark matter. However, its detection in the laboratory may be beyond current technology. The same could be said about dark electromagnetic waves, although they could well be the early dark radiation [57,58] proposed as one possible solution to the Hubble tension. Although the dark photon can be regarded as the sought-after dark matter, it is noteworthy that the associated DEM field exhibits MONDian characteristics. Therefore, we have a newly predicted fifth force that mimics dark matter, but not a new fermionic elementary particle as dark matter. From the point of view of fundamental physics, this difference (i.e., whether dark matter is fermionic or bosonic) is significant, as it determines whether there are only four fundamental forces or more than four.

**Author Contributions:** Conceptualisation, F.F., J.M.I., C.F.P. and T.P.S.; methodology, J.M.I. and T.P.S.; validation, F.F., J.M.I., C.F.P. and T.P.S.; formal analysis, J.M.I. and T.P.S.; investigation: F.F., J.M.I., C.F.P., T.P.S.; writing—original draft preparation, J.M.I. and T.P.S.; writing—review and editing, J.M.I. and T.P.S.; funding acquisition, F.F., J.M.I. and C.F.P. All authors have read and agreed to the published version of the manuscript.

**Funding:** J.M.I. was partially supported by FEDER/MCIN under grant PID2022-142407NB-I00 (Spain).

**Data Availability Statement:** All the data used in this article is already included in the article.

**Acknowledgments:** We thank Sukratu Barve, Kinjalk Lochan, and Cenalo Vaz for their helpful discussions. We thank the Vielberth Foundation for its support. C.P. thanks the Academic Research Sabbatical Program of the University of Regensburg for enabling this cooperation.

**Conflicts of Interest:** The authors declare no conflict of interest. The funders had no role in the design of the study; in the collection, analyses, or interpretation of data; in the writing of the manuscript; or in the decision to publish the results.

**Appendix A. MOND as Gravity in 2+1 Dimensions**

Any Riemannian manifold $\mathbb{M}$ has an associated Laplacian operator $\nabla^2$. The latter has a Green function $G(p, p')$ satisfying the Poisson equation $\nabla_p^2 G(p, p') = -\delta(p - p')/\sqrt{\det g_p}$. Thus, $G(p, p')$ can be regarded as the Newtonian potential created at point $p \in \mathbb{M}$ by a unit mass located at point $p' \in \mathbb{M}$. Now, $G(p, p')$ becomes singular as $p \to p'$. When $\mathbb{M}$ is two-dimensional, generally, we expect this singularity to be proportional to the logarithm of $d(p, p')$, the geodesic distance between the two points. Here, we prove this conjecture by explicitly computing the Laplacian Green functions for the two-dimensional sphere and the two-dimensional hyperbolic space.

*Appendix A.1. Green's Functions for the Laplacian in Two Dimensions*

Appendix A.1.1. The Plane $\mathbb{R}^2$

Given the Euclidean metric on the plane

$$\mathrm{d}s^2 = \mathrm{d}r^2 + r^2 \mathrm{d}\varphi^2, \tag{A1}$$

where $0 < r < \infty, 0 < \varphi < 2\pi$, the corresponding Laplacian operator reads

$$\nabla^2 = \frac{\partial^2}{\partial r^2} + \frac{1}{r}\frac{\partial}{\partial r} + \frac{1}{r^2}\frac{\partial^2}{\partial \varphi^2} \tag{A2}$$

The Green function $G(r, \varphi; r', \varphi')$ for a massless scalar on $\mathbb{R}^2$ satisfies the equation

$$\nabla^2 G(r, \varphi; r', \varphi') = \frac{-1}{r}\delta(r - r')\delta(\varphi - \varphi') \tag{A3}$$

Without loss of generality, we can assume $r' = 0$; this point is the origin. Moreover, by rotational symmetry, the Green function cannot depend on $\varphi$. We thus denote the Green function more simply by $G(r)$ and look for the solution to

$$\left(\frac{\mathrm{d}^2}{\mathrm{d}r^2} + \frac{1}{r}\frac{\mathrm{d}}{\mathrm{d}r}\right)G(r) = \frac{-1}{r}\delta(r) \tag{A4}$$

The solution reads

$$G(r) = [A - \Theta(r)]\ln r + B \tag{A5}$$

where $\Theta(r)$ is the Heaviside step function. We set $A = 0 = B$ and consider

$$G(r) = -\Theta(r)\ln r \tag{A6}$$

which, for $r > 0$, simplifies to

$$G(r) = -\ln r \tag{A7}$$

Since $r$ equals the geodesic distance between the origin and the point $(r, \varphi)$, our statement follows.

Appendix A.1.2. The Sphere $\mathbb{S}^2$

Given the standard round metric on the unit sphere $\mathbb{S}^2$,

$$\mathrm{d}s^2 = \mathrm{d}\theta^2 + \sin^2\theta \, \mathrm{d}\varphi^2, \tag{A8}$$

where $0 < \theta < \pi, 0 < \varphi < 2\pi$, and the corresponding Laplacian operator reads

$$\nabla^2 = \frac{\partial^2}{\partial \theta^2} + \cot\theta \frac{\partial}{\partial \theta} + \frac{1}{\sin^2\theta}\frac{\partial^2}{\partial \varphi^2} \tag{A9}$$

The Green function $G(\theta, \varphi; \theta', \varphi')$ for a massless scalar on $\mathbb{S}^2$ satisfies the equation

$$\nabla^2 G(\theta, \varphi; \theta', \varphi') = \frac{-1}{\sin \theta} \delta(\theta - \theta') \delta(\varphi - \varphi') \tag{A10}$$

Without loss of generality, we can assume $\theta' = 0$; this point is the north pole. Moreover, by rotational symmetry, the Green function cannot depend on $\varphi$. We thus denote the Green function more simply by $G(\theta)$ and look for the solution to

$$\left( \frac{\mathrm{d}^2}{\mathrm{d}\theta^2} + \cot \theta \, \frac{\mathrm{d}}{\mathrm{d}\theta} \right) G(\theta) = \frac{-1}{\sin \theta} \delta(\theta) \tag{A11}$$

The change of variables $x = \cos \theta$ reduces (A11) to

$$(1 - x^2) g''(x) - 2x g'(x) = \delta(x) \tag{A12}$$

where we set $g(x) = G(\theta)$. Equation (A12) is solved by

$$g(x) = \frac{1}{2} \ln \left( \frac{1 - x}{1 + x} \right) [A - \Theta(x)] + B \tag{A13}$$

where $A, B$ are arbitrary integration constants. We can set $A = B = 0$ to obtain

$$G(\theta) = \frac{1}{2} \ln \left( \frac{1 + \cos \theta}{1 - \cos \theta} \right) \Theta(\cos \theta) \tag{A14}$$

as the Green function for the Laplacian on the (upper hemisphere of the) 2-sphere, i.e., when $0 < \theta < \pi/2$; an analogous expression can be written for the lower hemisphere.

The point $\theta = 0$ is a singularity of $G(\theta)$. For $\theta \to 0^+$ we find

$$G(\theta) = -\ln \theta + \dots \tag{A15}$$

where the dots stand for regular terms in $\theta$. Now, in a small neighbourhood of the point considered, the tangent plane can be identified with the sphere. To this order of accuracy, the Green function (A15) exhibits the expected logarithmic singularity in the geodesic distance.

Appendix A.1.3. Hyperbolic Space $\mathbb{H}^2$

The metric on hyperbolic space $\mathbb{H}^2$ is

$$\mathrm{d}s^2 = \mathrm{d}\varrho^2 + \sinh^2 \varrho \, \mathrm{d}\varphi^2 \tag{A16}$$

where $0 < \varrho < \infty, 0 < \varphi < 2\pi$. The corresponding Laplacian operator reads

$$\nabla^2 = \frac{\partial^2}{\partial \varrho^2} + \coth \varrho \, \frac{\partial}{\partial \varrho} + \frac{1}{\sinh^2 \varrho} \frac{\partial^2}{\partial \varphi^2} \tag{A17}$$

The Green function $G(\varrho, \varphi; \varrho', \varphi')$ for a massless scalar on $\mathbb{H}^2$ satisfies the equation

$$\nabla^2 G(\varrho, \varphi; \varrho', \varphi') = \frac{-1}{\sinh \varrho} \delta(\varrho - \varrho') \delta(\varphi - \varphi') \tag{A18}$$

For the same reasons as above, it suffices to solve the ordinary differential equation

$$\left( \frac{\mathrm{d}^2}{\mathrm{d}\varrho^2} + \coth \varrho \, \frac{\mathrm{d}}{\mathrm{d}\varrho} \right) G(\varrho) = \frac{-1}{\sinh \varrho} \delta(\varrho) \tag{A19}$$

We apply the change of variables $x = \cosh \varrho$ to obtain

$$(x^2 - 1)g''(x) + 2xg'(x) = -\delta(x), \tag{A20}$$

where we set $g(x) = G(\varrho)$. Equation (A20) coincides with (A12); hence, an analogue of (A14) applies:

$$G(\varrho) = \frac{1}{2} \ln\left(\frac{1 + \cosh \varrho}{1 - \cosh \varrho}\right) \tag{A21}$$

Again, by Taylor-expanding around $\varrho = 0$ and dropping the irrelevant constants, we arrive at the expected logarithmic singularity in the geodesic distance:

$$G(\varrho) \simeq -\ln \varrho + \dots \tag{A22}$$

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
