# Peer review of "Theoretically Motivated Dark Electromagnetism as the Origin of Relativistic Modified Newtonian Dynamics"

_universe, doi:10.3390/universe10030123_

Round 1
Reviewer 1 Report
Comments and Suggestions for Authors
The paper, titled "Theoretically Motivated Dark Electromagnetism as the Origin
of Relativistic MOND," presents a captivating and innovative approach to understanding the relationship between general relativity (GR), relativistic MOND (RelMOND), and the broken electroweak symmetry. One of the paper's strengths lies in its bold exploration of connections between dark electromagnetism and the remaining unbroken symmetry after spontaneous symmetry breaking in the darkelectro-grav sector. The choice of the square-root of mass as the source charge for the dark electromagnetic force is intriguing, especially in light of the observed mass ratios of charged fermions.
Furthermore, the explanation of the dark electromagnetic force in the deep MOND regime, where acceleration is proportional to the square-root of mass, provides a compelling rationale for associating relativistic U(1)DEM gauge symmetry with the origin of MOND.
The section Introduction is too wide. I suggest the authors read again this section and try to shorten it.
The conclusion that dark electromagnetism serves as a mimicker of cold dark matter is particularly noteworthy. The paper asserts that the two are essentially indistinguishable in cosmological situations where cold dark matter successfully explains observations, such as CMB anisotropies and gravitational lensing. This conclusion opens new avenues for exploring the connection between dark electromagnetism and cold dark matter.
In summary, the paper offers a thought-provoking and well-reasoned exploration of the analogues between general relativity, relativistic MOND, and broken electroweak symmetry. The proposed framework introduces novel concepts, successfully linking disparate elements in a cohesive manner. This paper is a valuable contribution to the field and has the potential to stimulate further research and discussions in the scientific community.
Comments on the Quality of English Language
I think that the quality of English language is good.
Author Response
We thank the referee for their insights and encouraging comments.
Referee recommendation:
"The section Introduction is too wide. I suggest the authors read again this section and try to shorten it."
We agree with the referee. We have written a fresh, short introduction, which mainly lists the goal of the paper, and introduces the reader to the subsequent sections.
The major contents of the introduction have been moved to Section II titled `Theoretical origin of dark electromagnetism' which now reviews the theoretical background for this work.
We hope this rewriting of the paper will be found satisfactory by the reviewer. Thank you.
Reviewer 2 Report
Comments and Suggestions for Authors
This manuscript presents an ambitious theoretical framework that seeks to connect Modified Newtonian Dynamics (MOND) with dark electromagnetism, proposing an analogy between general relativity and relativistic MOND with broken electroweak symmetry. By introducing a unification model predicated on E8 × E8 symmetry, it incorporates dark electromagnetism and the entropic derivation of MOND, aiming to elucidate the understanding of galaxy rotation curves, dark matter, and the fundamental forces governing the universe. The paper meticulously explores the mathematical and theoretical underpinnings of this proposed unification and includes an extensive review section that provides valuable background information for the reader.
While the paper is intriguing and presents a novel approach, there are several areas where clarity and further elaboration are required:
(1) The manuscript posits that one of the E8 symmetries is spontaneously broken, leading to a dark electro-grav sector with a subsequent break in SU(2)R to accommodate classical gravity. This raises questions about the compatibility of a non-gravitational theory with standard big-bang cosmology, especially in relation to the formulation of dark electromagnetism and the pre-existing role of classical gravity. Furthermore, the action for darkelectromagnetism (72) is confusing; why classical gravity is already there?
(2) Dark matter is a cornerstone assumption in contemporary cosmology, supported by evidence from galaxy rotation curves, gravitational lensing of bullet cluster, CMB fluctuations and BAO, and the formation of large-scale structures. The manuscript would benefit from a more detailed discussion on how MOND, in its modified or unmodified forms, accounts for these phenomena, which is crucial for its viability as a comprehensive theory of the universe.
(3) The introduction of a characteristic acceleration (aM) related to the Hubble constant implies variability over time, suggesting that galaxies at different redshifts would exhibit distinct rotation curves. It would be advantageous for the manuscript to address observations supporting this hypothesis, as it is fundamental to the proposed framework.
(4) The derivation of dark matter density ratios to ordinary matter, from vector component contributions (lines 715 to 718) is assumed to be the same as the Coulomb part; this part requires a more robust justification. A more compelling argument in this section would strengthen the manuscript's claims.
(5) The characterization of the DEM field as analogous to cold dark matter, particularly as a source term in the Einstein equation, lacks clarity. Further explanation in this regard would enhance the manuscript's coherence.
(6) In Section 8.1, it is argued that critical acceleration and syummetry restoration. This story, symmetry restoration due to Unruh temperature, is subtle; I just leave some references, Unruh and Weiss (PRD29) and Hill (PLB155).
Addressing these points with detailed explanations or modifications could significantly enhance the manuscript's contribution to the field.
Additionally, the identification of typographic errors, such as those in Equation (72) and Reference 39, should be rectified in the final publication. Upon consideration of these revisions, the manuscript would be a strong candidate for publication in Universe.
Author Response
We sincerely thank the reviewer for their insightful comments which have led to an improvement in the manuscript.
Reviewer comments and author response:
While the paper is intriguing and presents a novel approach, there are several areas where clarity and further elaboration are required:
(1) The manuscript posits that one of the E8 symmetries is spontaneously broken, leading to a dark electro-grav sector with a subsequent break in SU(2)R to accommodate classical gravity. This raises questions about the compatibility of a non-gravitational theory with standard big-bang cosmology, especially in relation to the formulation of dark electromagnetism and the pre-existing role of classical gravity. Furthermore, the action for darkelectromagnetism (72) is confusing; why classical gravity is already there?
Author response: Thank you for this. We have added the following paragraph at the end of Section V :
A few further remarks about the proposed action principle in Eqn. (\ref{newactionp}) are in order. This form of the action is assumed to come into play after the electroweak symmetry breaking around a TeV scale. That is the epoch at which classical spacetime, general relativity and dark electromagnetism (i.e. relativistic MOND) emerge. This emergence is expected to give the same physical results as standard big bang cosmology, with cold dark matter exchanged for dark electromagnetism. Prior to this emergence, cosmology is governed by the unified $E_8 \times E_8$ theory, above the TeV scale. Since cosmological data are not yet available at such high energy scales, there is no contradiction between the octonionic theory and cosmology of the very early universe.
(2) Dark matter is a cornerstone assumption in contemporary cosmology, supported by evidence from galaxy rotation curves, gravitational lensing of bullet cluster, CMB fluctuations and BAO, and the formation of large-scale structures. The manuscript would benefit from a more detailed discussion on how MOND, in its modified or unmodified forms, accounts for these phenomena, which is crucial for its viability as a comprehensive theory of the universe.
Author response: We agree. We have added the following two paragraphs at the end of Section V :
Dark matter is a cornerstone assumption in contemporary cosmology, supported by evidence from galaxy rotation curves, gravitational lensing of bullet cluster, CMB fluctuations and baryon acoustic oscillations (BAO), and the formation of large-scale structures. Therefore it is important to assess how MOND fares vis a vis these aspects, as an alternative to dark matter. MOND does spectacularly well with galaxy rotation curves, being able to predict the rotation curve once the baryonic mass distribution of a galaxy is known from observations. In this regard it does better than the cold dark matter hypothesis, where a rotation curve has to be first known from observations and then a CDM distribution is assumed so as to fit the curve of velocities. MOND also provides a natural explanation for the Tully-Fisher relation, which is a challenge for CDM. MOND does not adequately account for cluster velocity dispersions, where CDM certainly does better. It has however been suggested that MOND's shortfall on cluster scales could be accounted for by the missing baryons which are currently unaccounted for in clusters.
CDM has a definite upper hand when it comes to formation of large scale structures, CMB anisotropies, BAO and gravitational lensing. Here one is in need of a convincing relativistic MOND which generalises general relativity at low accelerations and can then be convincingly applied in cosmology. The present proposal for dark electromagnetism is a step in that direction. For a detailed recent review of the status of MOND in astrophysics and cosmology, we refer the reader to Banik and Zhao \cite{BZ}.
(3) The introduction of a characteristic acceleration (aM) related to the Hubble constant implies variability over time, suggesting that galaxies at different redshifts would exhibit distinct rotation curves. It would be advantageous for the manuscript to address observations supporting this hypothesis, as it is fundamental to the proposed framework.
Author response: We agree. We have added the following paragraph above Eqn. (42):
The introduction of a characteristic acceleration ($a_M$) related to the Hubble constant implies variability over time, suggesting that galaxies at different redshifts would exhibit distinct rotation curves. This aspect has been discussed carefully for instance in the MOND review by Milgrom \cite{MilgromS} (see the subsection on `The significance of the MOND acceleration constant' therein, as well as the reference to \cite{Milgrom2017} where high redshift rotation curves are discussed in the context of MOND). Milgrom concludes that there are strong observational constraints on the variation of $a_M$ with cosmic time, and a value of $4a_M$ at $z \sim 2$ is essentially ruled out, hence excluding a dependence such as $a\propto (1+z)^{3/2}$.
(4) The derivation of dark matter density ratios to ordinary matter, from vector component contributions (lines 715 to 718) is assumed to be the same as the Coulomb part; this part requires a more robust justification. A more compelling argument in this section would strengthen the manuscript's claims.
Author response: We agree. The following remarks have been added in the relevant paragraph below Eqn. (49) :
The assumption that the vector components contribute in equal measure as the Coulomb part is a reasonable one because these considerations are being applied on the scale of the entire universe, including at high redshifts. Therefore relativistic motions must be taken into account, and as a result the `magnetic part' of the four-potential is expected to be as significant as the Coulomb part.
(5) The characterization of the DEM field as analogous to cold dark matter, particularly as a source term in the Einstein equation, lacks clarity. Further explanation in this regard would enhance the manuscript's coherence.
Author response: We agree. We have added the following remarks in the relevant paragraph below Eqn. (53):
The solution to this equation determines the DEM field, which then enters the Einstein equations as a source, as an alternative to dark matter. This source term is the potential energy of the DEM field, and it represents the enhanced gravitational interaction amongst baryons, instead of having to invoke dark matter to provide the sought for additional gravitational effects. The right hand side of Eqn. (\ref{45}) above is an illustration of this claim.
(6) In Section 8.1, it is argued that critical acceleration and syummetry restoration. This story, symmetry restoration due to Unruh temperature, is subtle; I just leave some references, Unruh and Weiss (PRD29) and Hill (PLB155).
Author response: Thank you for bringing these critical works to our attention. We have included these two references in the discussion on critical acceleration in Section 7, by adding the remark
`See however the critical analysis in this regard, by Unruh and Weiss \cite{Unruh} and by Hill \cite{Hill}.'
Addressing these points with detailed explanations or modifications could significantly enhance the manuscript's contribution to the field.
Additionally, the identification of typographic errors, such as those in Equation (72) and Reference 39, should be rectified in the final publication. Upon consideration of these revisions, the manuscript would be a strong candidate for publication in Universe.
Thank you. These have been corrected now.
Reviewer 3 Report
Comments and Suggestions for Authors
At the very beginning the authors note "The present paper is a modest attempt to initiate the research program outlined in this abstract" and, in fact, the paper should be considered as a proposition of further investigations. The program is based on some unification theory of fundamental forces given recently by P.Kaushik et al in ref.[1]. Employing this work the authors suggest that the relativistic version of modification of the Newtonian dynamics (RelMOND) has its origin in so called dark electromagnetism. Then in the rest of the paper they try to justify their idea. Of course this justification is not complete and hard work on the problem is needed. Nevertheless, since the proposition seems to be interesting, I recommend the reviewed work for publication as it stands now.
Author Response
We sincerely thank the referee for their supportive and encouraging comments on the manuscript.
Reviewer 4 Report
Comments and Suggestions for Authors
In the manuscript the authors try to give a theoretical explanation of the MOND suggestion by proposing that it is related to a dark electromagnetic background, presumably filling the entire Universe, but which the authors do not want to interpret as dark matter (but in Eq. (74) an extra matter like term still appears). The manuscript contains a lot of material, not always related to the topics. My first recommendation to the authors would be to significantly shorten their work, and concentrate on the relevant issues they would like to consider. Secondly, MOND faces a very serious problem when confronted with the observations, namely, the fact that the universal acceleration a_0 is not at all constant, but has very large variations from galaxy to galaxy-more exactly, it is not a universal constant. The same problem can be detected in the approach of the authors, which raises the same questions about the general validity of their model. This is an issue the authors must carefully discuss. But, as a theoretical speculation, the manuscript may still be considered for publication in Universe, once the above points are fully addressed.
Author Response
We thank the reviewer for their critical comments, which have helped improve the manuscript.
Reviewer suggestions and author response:
- The manuscript contains a lot of material, not always related to the topics. My first recommendation to the authors would be to significantly shorten their work, and concentrate on the relevant issues they would like to consider.
Author response: We agree with the reviewer. We also agree that the introduction was too long. Therefore we have written a short to the point introduction afresh. Most of the material from the introduction has been moved to a new section II, titled `Theoretical origin of dark electromagnetism'.
We have also removed the section `MOND as gravity in 2+1 dimensions' from the main body of the paper and placed it in an Appendix at the end of the paper. This considerably improves the logical flow of the paper. We thank the reviewer for suggesting that we remodel the paper.
2. Secondly, MOND faces a very serious problem when confronted with the observations, namely, the fact that the universal acceleration a_0 is not at all constant, but has very large variations from galaxy to galaxy-more exactly, it is not a universal constant. The same problem can be detected in the approach of the authors, which raises the same questions about the general validity of their model. This is an issue the authors must carefully discuss. But, as a theoretical speculation, the manuscript may still be considered for publication in Universe, once the above points are fully addressed.
We thank the reviewer for bringing this important point to our attention. We have added the following paragraph, and three references, at the end of Section III, so as to address this point.
"We also note that not all researchers are agreed on the presence of a universal critical acceleration scale in galaxies. In their analysis of 193 disk galaxies from the SPARC and THINGS databases, Rodrigues et al. conclude the absence of a fundamental acceleration scale in galaxies \cite{Rodrigues}. Whereas, in an accompanying paper Gaugh et al. perform a Bayesian analysis on galaxy rotation curves from the SPARC database and find strong evidence for a characteristic acceleration scale \cite{Gaugh}. See also the critical analysis in \cite{Chan} and references therein. Our outlook is that these analyses are a part of an ongoing debate, and for the purpose of our present theoretical discussion we will assume that a fundamental acceleration scale does exist on galactic scales."
Round 2
Reviewer 2 Report
Comments and Suggestions for Authors
I have reviewed the revised manuscript Universe-2824841 following the authors' response to my initial comments. I am pleased to report that the modifications and clarifications provided have substantially improved the clarity, coherence, and scientific rigor of the manuscript. The authors have addressed all the concerns raised in my previous review comprehensively, resulting in a manuscript that significantly contributes to the field of theoretical physics and cosmology.
The revised manuscript now includes a more detailed and scientifically fair comparison between Modified Newtonian Dynamics (MOND) in its relativistic form and the Cold Dark Matter (CDM) model. The authors have adeptly discussed how each model fares across various cosmological observations and phenomena, such as galaxy rotation curves, gravitational lensing, Cosmic Microwave Background (CMB) fluctuations, and baryon acoustic oscillations (BAO). This balanced discussion enables readers to weigh the merits and limitations of each model on solid theoretical and observational grounds.
Furthermore, the manuscript's exploration of dark electromagnetism as an emergent phenomenon post-electroweak symmetry breaking enriches the dialogue on alternative cosmological models. The addition of this discussion, alongside the careful consideration of the role of a characteristic acceleration scale related to the Hubble constant, provides a novel perspective on the evolution of cosmic structures and the nature of dark matter.
The authors have also effectively addressed the technical and conceptual questions regarding the formulation of dark electromagnetism, the symmetry breaking within the E8 framework, and the implications for classical gravity and cosmology. The clarifications and added references enrich the manuscript's theoretical foundation and contextualize its contributions within the broader scientific discourse.
In light of the substantial revisions and the thorough response to the initial critique, I recommend the manuscript for publication in Universe. The work presents a compelling and well-argued case for considering alternative frameworks in cosmology, contributing valuable insights to ongoing debates about the nature of dark matter and the dynamics of the universe.
Reviewer 4 Report
Comments and Suggestions for Authors
The authors have improved their manuscript, and hence I think the present version is suitable for publication in Universe.
Comments on the Quality of English LanguageQuality of English good.